psychology

cognitive modelling, reproducibility, open science, preregistration, transparency

**Author for correspondence:**
Sophia Crüwell
e-mail: sophia.cruewell@charite.de

# Preregistration in diverse contexts: a preregistration template for the application of cognitive models

## Sophia Crüwell[1,2,3] and Nathan J. Evans[2,4,5]

[1]Meta-Research Innovation Center Berlin (METRIC-B), QUEST Center for Transforming Biomedical Research, Berlin Institute of Health, Charité – Universitätsmedizin Berlin, Berlin, Germany
[2]Department of Psychology, University of Amsterdam, Amsterdam, The Netherlands
[3]Department of History and Philosophy of Science, University of Cambridge, Cambridge, UK
[4]School of Psychology, University of Queensland, Queensland, Australia
[5]School of Psychology, University of Newcastle, Callaghan, Australia

SC, 0000-0003-4178-5820

In recent years, open science practices have become increasingly popular in psychology and related sciences. These practices aim to increase rigour and transparency in science as a potential response to the challenges posed by the replication crisis. Many of these reforms—including the increasingly used *preregistration*—have been designed for purely experimental work that tests straightforward hypotheses with standard inferential statistical analyses, such as assessing whether an experimental manipulation has an effect on a variable of interest. But psychology is a diverse field of research. The somewhat narrow focus of the prevalent discussions surrounding and templates for preregistration has led to debates on how appropriate these reforms are for areas of research with more diverse hypotheses and more intricate methods of analysis, such as cognitive modelling research within mathematical psychology. Our article attempts to bridge the gap between open science and mathematical psychology, focusing on the type of cognitive modelling that Crüwell *et al.* (Crüwell S, Stefan AM, Evans NJ. 2019 Robust standards in cognitive science. *Comput. Brain Behav.* **2**, 255–265) labelled *model application*, where researchers apply a cognitive model as a *measurement tool* to test hypotheses about parameters of the cognitive model. Specifically, we (i) discuss several potential researcher degrees of freedom within model application, (ii) provide the first preregistration template for model application and (iii) provide an example of a preregistered model application using our preregistration template. More broadly, we hope that our discussions and concrete proposals constructively advance the mostly abstract current debate

surrounding preregistration in cognitive modelling, and provide a guide for how preregistration templates may be developed in other diverse or intricate research contexts.

# 1. Introduction

The replication crisis has been an issue for psychology and related fields since at least 2011 [1]. Scholarship surrounding issues likely underlying this crisis dates back much further (cf. e.g. [2–4]). In response to these issues, and in an attempt to increase rigour and replicability, researchers have proposed a variety of reforms—often termed *open science* practices—which emphasize rigour, specificity, the constraint of flexibility, and transparency. These practices include data sharing [5], preregistration [6] and the journal article format Registered Reports [7]. The term *open science* is commonly used to refer to these practices (and other practices promoting transparency and access) as they encourage openness in the sense of transparent and accessible research [8], which in combination with specificity and constraint are seen by many as essential to counteract the effect of cognitive biases and other pressures that may influence scientific findings [9]. The current article focuses on the open science practice of preregistration, which we discuss in more detail below, and how it might be implemented for model application (more on this in the subsection on cognitive modelling; [10]) in cognitive modelling studies within mathematical psychology.[1] The goal of this paper is twofold: (i) to introduce a template that modellers can critique, improve upon, and use if they want to; (ii) to show Open Science advocates another extension of their existing templates and how the principles they try to further may also be applied in model application. We realize that there is a disconnect between the two communities: Open Science advocates believe that their Open Science principles are broadly applicable, whereas modellers by and large believe that these principles are not applicable to cognitive modelling. Overall, this debate has been held in the abstract, which can be useful, but should be complemented with concrete proposals. Our template for preregistration in model application is one such concrete proposal.

# 2. Preregistration

Preregistration involves the specification of a researcher's plans for a study, including hypotheses and analyses, typically *before* the study is conducted. This usually takes the form of a time-stamped document that contains these plans, which is made available online. It has been suggested that preregistration can help to make the research process more transparent, to constrain researcher degrees of freedom (i.e. undisclosed flexibility in study design, data collection, and/or data analysis. More on this below in the section 'Researcher Degrees of Freedom in Model Application'; Simmons *et al.* [11]), and to help alleviate the effects of questionable research practices (QRPs) such as hypothesizing after results are known (HARKing; [12]) or p-hacking [6,9,13]. This is important as each of these practices can render inference based on seemingly confirmatory analyses unwarranted ([6,13]; though see also Szollosi & Donkin [14] for a discussion on the potentially problematic dichotomy between confirmatory and exploratory analyses).[2] Thus, while most published studies in psychology appear to be confirmatory, it may be difficult to know whether these studies truly are confirmatory without the *a priori* specification of hypotheses and analysis plans, particularly given the incredibly high incidence of findings falling in line with the 'confirmatory' predictions of psychology studies [16]. Preregistration may help researchers distinguish *a priori* predictions (and even justifications of e.g. analysis choices; more on this below in 'Motivations and justification') from *post hoc* explanations. This is not only helpful for the researchers themselves but may also serve as a rudimentary but credible signal of accurate prediction for readers not familiar with a field or theory, which may be particularly

---

[1]While we think that this work may be applicable to *model application* studies (more on this below) across cognitive modelling, the main focus of this paper is cognitive models within mathematical psychology. Our proposed template may not be well or immediately suited for cognitive models in other areas, but we believe that it may also serve as a useful starting point there.

[2]More specifically, in discussions surrounding preregistration, research is often dichotomized into strictly exploratory and strictly confirmatory research. 'Confirmatory' research refers to research where hypotheses and analyses were planned before the start of the study. However, this dichotomy is not necessarily a useful one [14,15], and exploratory and confirmatory research are likely part of a broader spectrum. Nevertheless, we will use these terms in the context of describing the debate surrounding preregistration in cognitive modelling to reflect the language and concepts used by both proponents and critics.

useful for niche and intricate theories and areas of research.[3] At the very least, even a basic preregistration will give the reader further context on the research process. Furthermore, preregistration can have other important benefits for the scientific process, such as helping to organize and streamline the research workflow, and providing researchers with the opportunity to flesh out, critique, and alter their design choices to ensure that they properly test their *a priori* predictions.

The fact that a study is preregistered should not be taken as a marker of quality by itself, as the preregistration document may lack the specificity needed to effectively constrain potential researcher degrees of freedom [17], and the decisions made in the preregistration may not be well justified or appropriate [18]. The Registered Reports format allows for an assessment of the quality of the pre-specified plan through an initial round of peer-review before the study is conducted, meaning that researchers can alter their pre-specified plans based on reviewer feedback [7]. This is not the case, however, for the standard practice of preregistration, and many psychology journals do not currently include a Registered Report article format, meaning that researchers may initially struggle to create preregistration documents that are appropriately detailed and justified [13]. Several preregistration templates have been developed to assist researchers in creating preregistration documents, such as those provided by the Open Science Framework (OSF; https://osf.io/prereg/) and AsPredicted (https://aspredicted.org/), as well as checklists to assess the quality and constraint of a preregistration document [19]. These initial templates and checklists were originally designed as general-purpose tools for experimental psychology.[4] Therefore, they are applicable to studies where researchers are interested in testing straightforward hypotheses, such as whether an experimental manipulation has an effect on a variable of interest, with standard analysis tools, such as a null hypothesis significance test on an interaction term within an ANOVA.

A large proportion of psychology studies fall within the standard experimental framework that these general-purpose templates and checklists have been designed to accommodate, making these tools of broad use to many researchers in psychology. But psychology is a diverse field of research, and several areas of psychology commonly involve studies with more diverse hypotheses and more intricate methods of analysis. Importantly, the central focus of preregistration endeavours on purely experimental research[5] has led to debates on how appropriate preregistration is for psychological research that is not purely experimental. In particular, this debate has frequently occurred in and surrounding the area of *cognitive modelling*, where researchers use mathematical (or computational) models that are formal representations of cognitive processes to better understand human cognition [10,18,22–26]. Some question the general usefulness of preregistration in areas of psychological research with more diverse hypotheses and more intricate analyses, e.g. pointing out the exploratory nature of model *development* or focussing on the development of strong theory ([18,25]; see also the 'Cognitive Modelling' and 'Issues and Peculiarities in Preregistering Model Application' sections where we address several common challenges and misconceptions). Others believe that preregistration could still serve an important purpose in constraining researcher degrees of freedom in other categories of cognitive modelling, and even where there is strong theory [10,22,24]. However, the preregistration tools currently available to researchers may make achieving proper constraint practically infeasible, as the exact researcher degrees of freedom in these areas of research can differ greatly from those in purely experimental psychology ([10,22,24,26]; though see also [27] for a cognitive modelling study with a well-constrained preregistration using existing tools). Recent research has already begun to create more specific preregistration templates for more specific areas of research, such as in qualitative research [21,28], experience sampling methodology [29], secondary data analysis [30–32] and fMRI studies [20]. Further development of method- and field-specific preregistration templates and checklists may improve the applicability of preregistration to areas of psychology research with more diverse hypotheses and more intricate analyses, similar to how the development of general-purpose, structured preregistration templates and checklists have been suggested to help researchers create constrained preregistration documents for purely experimental studies [17].

Our article aims to bridge the gap between previous preregistration endeavours and research in areas of psychology with diverse hypotheses and diverse analyses. At a general level, we wish to showcase that preregistration can be used to constrain potential researcher degrees of freedom in more diverse research

---

[3]We thank Alex Holcombe for alerting us to this excellent potential advantage of preregistration.

[4]Note that these initial templates have now been extended to more specific templates for some areas of research, such as fMRI studies [20] and qualitative research [21], which we discuss in more detail later.

[5]Note that while mathematical modelling research often involves experimental tests, when we talk of (purely) experimental research here and below, we refer to experimental research that uses standard inferential statistical methodology.

contexts, and that field-specific preregistration templates are important for both understanding and constraining these degrees of freedom. More specifically, we develop an initial preregistration template for cognitive modelling research, building on the initial suggestions of Lee *et al.* [24], who proposed the concept of 'registered modelling reports', as well as Crüwell *et al.* [10], who proposed four potential categories for different types of cognitive modelling and suggested that different categories are likely to have different researcher degrees of freedom (i.e. different categories require different templates). Our article focuses on the category of cognitive modelling that Crüwell *et al.* [10] labelled *model application*, where researchers apply a cognitive model as a *measurement tool* to test hypotheses about parameters of the cognitive model, such as assessing whether an experimental manipulation has an effect on a parameter of interest.

The remainder of this article will take the following format. First, we explain the concept of cognitive modelling, as well as the category of model application and how it differs from other categories of cognitive modelling. Second, we detail several additional researcher degrees of freedom that we believe are relevant to model application, and how they can make the preregistration process more complicated than in purely experimental work. Third, we provide a preregistration template for model application, including an example implementation to showcase how it can help to constrain researcher degrees of freedom. Fourth, we discuss the potential limitations of our template—most notably, that our template only covers model application, and that cognitive modelling studies often involve an interplay between different categories of cognitive modelling—and future directions for developing templates for other categories of cognitive modelling. We hope that our discussions and proposals will constructively advance the debate surrounding preregistration in cognitive modelling, and provide a guide for how preregistration templates may be developed in other diverse or intricate research contexts.

## 2.1. Cognitive modelling

Cognitive modelling is the formal description of theories about psychological processes [33]. Unlike statistical models, cognitive models contain parameters that directly reflect psychological constructs [34], and the assumptions of cognitive models are designed to align with actual human behaviour. Cognitive models do not intend to provide perfect representations of psychological processes, but rather useful formal instantiations of psychological theory (e.g. 'All models are wrong, but some are useful.'; [35], p. 202). This can enable more precise insight than is possible using verbal theories [36]. Importantly, 'cognitive modelling' serves as an umbrella term that comprises diverse hypotheses and methods of analysis, with cognitive models being capable of serving a variety of different functions to answer a range of different research questions, spanning the spectrum from purely exploratory to purely confirmatory.

Although the diversity within cognitive modelling research is undoubtedly a positive feature, as it allows for cognitive models to provide unique insights into psychological processes in a range of different contexts, this diversity can also lead to some potential pitfalls. One key example is the increasingly discussed abundance of 'modeller's degrees of freedom' [37,38], where differences in approaches and interpretations—for example, in interpreting model success and model failure [36,39]—can influence the results and conclusions of a study (though freedom in modelling is not always viewed as a negative, e.g. [25]). Furthermore, the potential degrees of freedom in approaches and interpretations are not always transparent, which can impede the effectiveness of the research process [40]. Importantly, these issues are precisely what many open science practices—particularly preregistration—have been designed to address in purely experimental areas of psychological research. Therefore, it seems possible that preregistration could be a useful tool for at least some parts of cognitive modelling research, reducing the potential modeller's degrees of freedom and allowing for a more transparent research process.

Despite the intuitive appeal of preregistration as a possible solution to the abundance of modeller's degrees of freedom within cognitive modelling, the potential introduction of preregistration to cognitive modelling has been a contentious issue [10,18,22–26]. One possible reason for this debate is the previously mentioned diversity within cognitive modelling, where cognitive models can be used in a range of different contexts to answer a range of different research questions. Importantly, putting a single set of constraints on cognitive modelling across all contexts would greatly reduce the modeller's degrees of freedom, but would also prevent much of the diversity that makes cognitive models such a useful instrument within psychological research. Different contexts are also likely to have different relevant degrees of freedom, meaning that a single set of constraints for the entirety of cognitive modelling would likely be inappropriate and ineffective. As a potential first step to allowing the constraint of modeller's degrees of freedom, while also maintaining the diversity of cognitive

modelling, Crüwell *et al.* [10] proposed that it may be useful to separate cognitive modelling research into four discrete categories: *model development*, *model evaluation*, *model comparison* and *model application*. Model development involves the initial development of a model, or the extension/reduction of an existing model to create a new model, which is often an iterative, exploratory process and thus not well suited to preregistration (for a discussion on how open theorizing can help constrain inference, see [41]). Model evaluation involves assessing whether a model, or multiple models, can qualitatively capture certain trends in empirical data, with these trends often referred to as 'qualitative benchmarks'. Model comparison involves directly contrasting multiple models on their ability to account for a set of empirical data, which is usually performed quantitatively through model selection methods (e.g. AIC, [42]; BIC, [43]; see [44] for a discussion). These latter two categories may also benefit from preregistration (though see [45] for an alternative approach for model comparison: *formal* model comparison). Also note that while model application and model comparison can both make use of model selection methods, they are separated by their intended purpose. While model application aims to determine whether parameters differ between conditions or groups within a model, model comparison aims to quantitatively compare theoretically distinct models, where the models differ in at least some aspect of the proposed underlying process. Furthermore, it should be noted that assessing the descriptive adequacy of a model—i.e. performing model evaluation to ensure that the model provides a good account of the trends in the data—is also often a part of studies that implement model application (see the 'Limitations' section for a more detailed discussion).

Model application, which will be the focus of our article, involves using an existing cognitive model to answer research questions about specific components of the underlying cognitive process—note again that it is the intended purpose of using the model that determines the category of cognitive modelling research. More specifically, in model application the chosen model is assumed to provide an accurate representation of the cognitive process underlying the data, and researchers are often interested in how the components change over experimental conditions and/or groups, with these changes often having meaningful interpretations for related theories (e.g. task differences between older and younger adults; [46]). Model application involves using cognitive models in a similar manner to statistical models (e.g. ANOVA), but with the assessments performed on the theoretically meaningful parameters estimated within the cognitive model rather than on the variables directly observed within the data. This difference creates several researcher degrees of freedom that are not present in purely experimental research using only statistical models. Nevertheless, model *application* is the category of cognitive modelling that is most closely related to the purely experimental research that previous preregistration efforts have been focused on—particularly its highly confirmatory nature makes model application the ideal category for initial preregistration efforts within cognitive modelling. In the next section, we will provide a concrete discussion of potential modeller's degrees of freedom within model application—degrees of freedom that are not present in purely experimental research—which will form the basis for our preregistration template.

# 3. Researcher degrees of freedom in model application

What makes a good preregistration, and thus a good preregistration template? There are a variety of proposals for high-quality preregistrations, which differ in the amount of detail required from the researcher regarding their exact research plan (e.g. [19,47,48]). Similarly, different preregistration templates ask different questions, which differ in their level of detail and how much they prompt the researcher with specific instructions. Broadly, a good preregistration should make it easier to differentiate between exploratory and confirmatory research, by making transparent the choices made in a research project over time and thus transparently constraining researcher degrees of freedom (i.e. flexibilities in study design, data collection, analysis and reporting; [11,19]). The intended use of preregistration is not limited to potentially constraining the effects of common cognitive biases—preregistration may also help to organize and streamline a research workflow. Previous research has suggested that preregistration templates can have meaningful differences on the specificity of preregistrations. For example, a comparison between preregistration documents that used one of the two commonly used OSF preregistration templates showed that the less detailed OSF Standard Pre-Data Collection Registration template (since replaced by an improved standard form) led to less specific preregistrations than the more detailed OSF Prereg Challenge template, with the latter being better at restricting researcher degrees of freedom [17].

Bakker *et al.* [17] also proposed a stringent checklist for assessing the quality of preregistrations by checking whether all applicable researcher degrees of freedom are appropriately restricted by the

preregistration. On a general level, it has been emphasized that a preregistration should lead to a plan that is specific (all steps to be taken are included), precise (each step is unambiguous), and exhaustive (there is no room left for other steps to be taken), so as to restrict the use of researcher degrees of freedom [19]. Using standard templates, researchers in purely experimental areas of psychology may more easily create a specific, precise, and exhaustive preregistration that restricts the researcher degrees of freedom present in these experimental contexts. But restricting these commonly discussed generic researcher degrees of freedom is not sufficient for a successful preregistration in more diverse contexts, such as model application. In fact, there have been cognitive modelling preregistrations submitted to the OSF that mentioned no more than the general modelling approach,[6] which is understandable given that these preregistration templates were not designed for cognitive modelling studies.

In order to appropriately apply preregistration to research contexts that are not purely experimental, it is important to identify the unique researcher degrees of freedom within the area of research that the preregistration should ideally constrain. Below, we identify several degrees of freedom that we believe are present within model application, and then provide a more detailed discussion of these degrees of freedom and why they are important. We devised this non-exhaustive list qualitatively by reviewing the literature and using our research experience to propose and discuss candidate researcher degrees of freedom, similarly to Wicherts *et al.* [19]. Our discussions of possible researcher degrees of freedom are influenced by previous research on robust practices in cognitive modelling (e.g. [10,18,24–26,49]), blinded inference (e.g. [37,38]), previous preregistration attempts in cognitive modelling (e.g. [27]), and the vast literature of model application work with models of human response time (e.g. [46,50–58]). Note that many previously identified researcher degrees of freedom—such as studying a vague hypothesis, *ad hoc* exclusion of outliers, or specifying the pre-processing of data in an *ad hoc* manner [11,19]—are also applicable to model application, meaning that previous lists of researcher degrees of freedom should be *extended* for model application by adding the following category-appropriate modeller's degrees of freedom:

---

**Cognitive model**
M1: Choosing a type of cognitive model.
M2: Specifying the exact parameterization of the model(s).
M3: If applicable, specifying the (theoretical) motivation for these choices.
**Parameter estimation**
E1: Deciding on the method of parameter estimation.
E2: Specifying settings/priors for parameter estimation.[7]
E3: If the data are going to be summarized into descriptive statistics, specifying which
    descriptive statistics will be used and how.
**Statistical inference**
I1: Choosing a method of statistical inference on parameters (e.g. when comparing conditions).
    (This is largely covered by existing templates)
I2: Specifying what parameters will be assessed (e.g. allowed to vary across experimental conditions).
**Robustness checks**
RC1: Performing and/or reporting robustness checks.

---

## 3.1. Issues and peculiarities in preregistering model application

The concrete list of modeller's degrees of freedom above provides an indication of what factors should be constrained by an exhaustive preregistration in model application, beyond the general researcher degrees of freedom. Here, we discuss these factors and some potential pitfalls surrounding them, including the pre-specification of theoretical motivations, model parameterizations, estimation methods, preexisting data, outlier exclusions, parameter recovery studies, and robustness checks, as well as alternative minimal standards (i.e. when preregistration is not possible or not desired) for transparency in model application.

---

[6]Note that we do not cite specific example here, as we do not wish to target specific researchers who have made good-faith attempts to preregister their cognitive modelling studies. Nevertheless, a quick search on most registration platforms—such as the OSF registries—provides a demonstration of the wide-ranging preregistration practices for cognitive modelling studies.

[7]However, note that there are arguments for why the priors should be considered as part of the model itself, rather than part of the estimation method [59,60]. Our choice to place the priors in the estimation section is not intended as a stance on this theoretical issue, but rather to make our template as applicable to both Bayesians and frequentists as possible.

### 3.1.1. Motivations and justifications

First and foremost, it is important that the choices made at all stages of a model application study are justified and motivated, which is often not the case in cognitive modelling studies that use model application [37,38]. The supplementary materials of Dutilh *et al.* [37] contain a collection of analysis decisions and motivations of different teams analysing the same datasets using different methods.[8] While many of the conclusions made by different teams were similar, the discrepancies caused by each team's unique choices reveal how modeller's degrees of freedom can influence inferences, making well-motivated choices crucial. Considering that one of the main advantages of cognitive modelling is generalizability based on substantive explanations for psychological phenomena [34], one could argue that it is also a modeller's degree of freedom to not prespecify motivations or justifications for different decisions. In the context of model evaluation, failing to specify which assumptions are central theoretical components of a model (i.e. the *core* assumptions) and which are only included out of practical necessity (i.e. the *ancillary* assumptions) can lead to situations where core assumptions can be written off as ancillary assumptions when they perform poorly, or do not produce the desired results (see [61], for a discussion). In model application, a lack of specification and motivation may allow a researcher to change the form of a model depending on whether the model applied to their data results in the difference between groups they were looking for. Similarly, when an assumption is not theoretically motivated, it should be clearly noted to avoid confusion or overinterpretation [62]. Therefore, preregistration seems to be useful both for the *a priori* justification of choices, and for the clarification of which choices do not have clear justifications.

### 3.1.2. Model parameterizations

It is crucial to clearly pre-specify model(s) of interest—'the players of the game' [24]—as well as the specific parameterization(s) of each model of interest, as there are a large number of possible variants of any model that may be applied [63]. Note that when we refer to the 'model parameterization', we are referring to both the different potential variants within a single model (e.g. a diffusion model with or without between-trial variability in drift rate), as well as the different ways that the parameters can be expressed within the same variant of a model (e.g. a diffusion model with starting point estimated as an absolute value or as a value relative to the threshold). We believe that both of these aspects are important to specify within the preregistration, as variations in how the parameters are expressed could potentially lead to different results (e.g. due to changes in priors, which can influence Bayesian model selection [59,64]). An ideal preregistration would include a complete plate diagram (or equivalent), such as figure 1 below, where the model and parameterization is specified, as well as any explicit dependencies between parameters, and any restrictions placed on the values of the parameters (see appendix A). For example, if Bayesian hierarchical modelling is used for parameter estimation, the structure of the hierarchical model and the prior distributions over the parameters should be included within the parameterization. Any *post hoc* addition to, or modulation of, the model or the parameterization should be clearly labelled as exploratory rather than confirmatory. Otherwise, the ability to adjust these factors can—at least in some cases—allow cognitive models to produce any possible result (cf. [65,66]), which would arguably be the ultimate degree of freedom. An effective preregistration of the model and the model parameterization can constrain these potential researcher degrees of freedom, which would help ensure more rigorous model application work.

### 3.1.3. Parameter estimation

Following the model and its parameterization, the next subsection of the preregistration template for model application is concerned with parameter estimation. In any situation where parameters are estimated, researchers should be clear on how they will estimate the parameters, and how the estimated parameters will be used as part of the analyses. In many cases, however, researchers might only be interested in parameter estimation as a means of getting to the inference measure, as is the case for many Bayes factor approximation methods [64,67–69], which should be explicitly stated. Moreover, it is not always necessary to estimate the parameters to obtain an inference measure (see [70]), and in these cases the parameter estimation section may be replaced with a statement that

---

[8]These electronic supplementary material can be found at http://osf.io/egrnn.

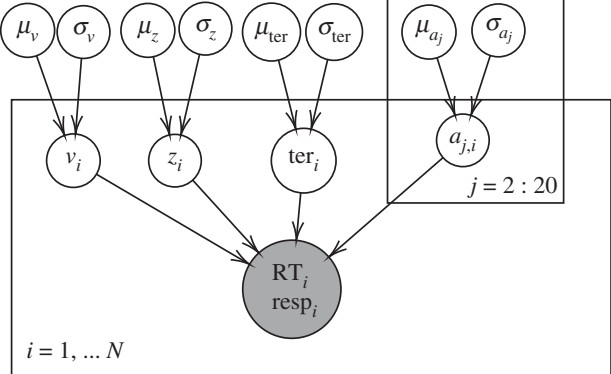

**Figure 1.** Example of a plate diagram used to preregister the parameterization of the model, taken from our example application. In this diagram, $i$ indexes participants and $j$ indexes blocks. The corresponding distributions can be found in the full preregistration in appendix A.

explains why parameter estimation is not of interest here, meaning that any subsequent estimation of parameters would belong in an exploratory section.

### 3.1.4. Secondary data analysis

One part of standard preregistration documents that will likely need to be adapted for many model application studies is information about original data collection, as many cognitive modelling studies reuse existing data. Preregistration of secondary data analyses is a challenge in itself, which has begun to be addressed with several proposed templates (see [30–32]). Although such a preregistration will not function in quite the same way as it is meant to with original data, the additional transparency of openly stating all preexisting knowledge of the specific secondary datasets that one plans to analyse will hopefully provide similar added credibility, and at the very least provide the reader with more context on the research process that produced the results presented in the corresponding paper [13]. Therefore, our preregistration template integrates parts of the preregistration template for secondary data proposed by Weston *et al.* [31]. In cases where existing datasets are used, the sections on conducting and designing the study—'Sampling Plan', 'Design Plan' and 'Variables'—should be replaced by the section 'Data Description for Preexisting Data'. Note that the preregistration of secondary data analyses makes it particularly difficult to verify whether the preregistration was completed before the researchers had started exploring the data. For example, a 'postregistration'— which details analyses that have already been performed [24]—could be easily and fraudulently disguised as a 'preregistration of existing data'. Nonetheless, we believe that the goal of preregistration is not to prevent dishonesty or outright fraud, but to help researchers avoid fooling themselves into believing that their *post hoc* explanations and explorations were *a priori* predictions and decisions.[9] More generally, we believe that attempting to eliminate dishonesty and fraud is beyond the scope of typical preregistration efforts, and that other methods would be required to verify the sequence of events (e.g. [73]).

### 3.1.5. Outliers and power analysis

There are two aspects already included in general preregistration templates that still deserve a separate discussion in the context of cognitive modelling in general and model application in particular. Firstly, a necessary pre-specification that is already included in most preregistration templates is the decision on how outliers are excluded and data are pre-processed. In purely experimental studies, this is often focused at the level of entire participants. This becomes more intricate in model application studies, as models are often estimated using the combined information from individual trials, meaning that researchers may want to exclude individual trials as well as entire participants from data analysis. Consequently, researchers conducting model application studies should clearly specify exclusion

---

[9]Cf. the debate in philosophy of science surrounding prediction and accommodation of evidence. The overall consensus of this debate is that accommodated evidence/postdiction can *sometimes* be epistemologically equivalent to prediction ([71]; for empirical evidence supporting this consensus: [72]).

criteria for both individual trials and entire participants, and in which order these exclusions should be applied, as the exclusion of a subset of trials could influence whether or not a participant is excluded by the participant exclusion criteria. Secondly, in purely experimental studies, the aspect of power analysis is usually focused on the classic concept of statistical power (i.e. the probability of rejecting the null hypothesis given that it is false, within a null hypothesis significance testing framework). This also becomes more intricate in model application, as the analysis process is not simply a statistical test, and studies often use inference techniques other than null hypothesis significance testing. Other factors in model application are highly dependent on the experimental design, however, such as the precision of the parameters estimated from the cognitive model. Therefore, when considering preregistration in model application, the classical power analysis used within purely experimental contexts could potentially be replaced with a parameter recovery study that matches the preregistered experimental design, model, and model parameterization [63,74]. More specifically, parameter recovery studies involve generating simulated datasets from a model using different sets of parameter values, fitting the same model to these datasets, and assessing whether the estimated parameter values match the generating parameter values. Importantly, parameter recovery studies assess the reliability of the measurement properties of the model—which can differ within a model based on experimental design and model parameterization [74]—and models with poor recovery can provide spurious conclusions. Similar to power analyses, researchers could use the results of parameter recovery assessments (or in the case of model selection approaches to model application, model recovery assessments; see [44] for a discussion) to design their experiment and determine their sample size, choosing a design, a number of participants, and a number of trials that provide robust parameter recovery. Furthermore, as models are often thought to be crude approximations of actual cognitive processes, researchers could assess how robust their assessment is to model misspecification by generating data with a range of different models and assessing how well similar constructs are recovered in their chosen model (e.g. a cross-fit assessment; [74,75]). We believe that the specification of precise parameter recovery standards is beyond the scope of the current article, though we do believe that researchers should strongly consider performing some type of parameter recovery and including it in their preregistration.

### 3.1.6. Robustness checks

Although we believe that preregistration shows clear promise as a useful tool within model application, we can understand how some researchers within the field of mathematical psychology may still be skeptical regarding its utility. One previous argument against the utility of preregistration has been that cognitive modelling research has other, superior practices for ensuring the robustness of their findings, such as robustness checks (see e.g. a tweet by Morey [76]); robustness checks are similar to what has recently been referred to as 'multiverse analyses' in the context of psychological research [77]. We disagree with the stance that robustness checks provide an alternative to preregistration, as the performing and reporting of robustness checks provide a large potential researcher degree of freedom. Specifically, if researchers were to only perform or report the robustness checks that were successful in showing their results to be robust, then readers would likely become overconfident in the results of the study, as the findings were robust against all *reported* robustness checks. Thus, although we agree that robustness checks are an important part of model application (and more generally, cognitive modelling research), we disagree that they are an alternative to preregistration. Instead, we believe they are an aspect of the study that should be included *within* a preregistration document.

### 3.1.7. Well-documented code

There may be other good arguments against the use of preregistration in model application and researchers might be interested in alternative methods that also provide increased transparency. Any researcher looking for an alternative to preregistration could consider the provision of well-documented code as a minimal standard for making their study more transparent. In fact, writing and posting well-documented analysis code on an online repository before the analysis is conducted could even be considered a minimal preregistration, as this could function to constrain many of the degrees of freedom that we discussed above. But it should be noted that without sufficient documentation and comments the analysis code may be difficult to use, and thus, the results may be difficult to reproduce. Furthermore, the documentation would have to be much more extensive than standard code documentation, as it would also need to include information on the theory behind the model to enable meaningful interpretation [62]. Therefore, researchers who choose this alternative to

preregistration should invest a reasonable amount of time into developing well-documented code. Well-documented code can also play a complementary role within preregistration (and for open modelling in general, see e.g. [78] for a discussion of distributed collaboration). In general, we believe that well-documented code should be provided whenever possible, as it can potentially help to make even the most constrained preregistration document even more transparent. Nevertheless, we do not believe that well-documented code should be a separate section within a preregistration document; rather, we believe that well-document code may be used to complement each section of the preregistration document, supporting what will be done with the code that will be used to do it.

# 4. A template for preregistration in model application

Taking the previous considerations into account, we developed a template for preregistration in model application, which can either be used in the context of standard preregistration, or as a basis for the Registered Modelling Reports journal format suggested by Lee *et al.* [24], which builds on the conventional Registered Reports journal format [7]. Note that we are *not* attempting to claim that this is the only way that researchers should preregister studies involving model application, or that researchers must use preregistration in model application studies. Rather, we believe that our preregistration template *may* be a useful tool to help ensure robustness and transparency in model application studies; whether this is the case is ultimately an empirical question. It should also be noted that there is already at least one example of a modelling study using a detailed preregistration [27], meaning that our template is not the only method for creating a highly constraining preregistration document in model application. Unfortunately, this study by Arnold *et al.* [27] seems to be the exception to the rule: preregistration in model application appears to be quite rare, and most other preregistration documents in model application do not sufficiently constrain the types of modeller's degrees of freedom that we discuss above. This suggests that a preregistration template specific to model application and constraining the appropriate modeller's degrees of freedom could be of great use.

To make our preregistration template as concrete as possible, each part of our preregistration template will be accompanied by an example related to our example application. The details of our example application can be found in the next section. Note that we do *not* intend for our example application to be seen as a perfect or exemplary instance of preregistration in model application. Rather, our example application is intended to serve as our realistic attempt to preregister a research question involving model application within the area of reward rate optimality using our proposed preregistration template. Importantly, we do not believe that preregistration is a 'one size fits all' solution: the preregistration of different model application studies (or even different analyses of the same study) are likely to differ from one another in many respects, meaning that the concept of a perfect or exemplary preregistration may not be particularly helpful. However, we hope that our example application serves as a useful example for readers who wish to use our template to preregister their own model application studies.

## 4.1. Preregistration template

Taking into account the modeller's degrees of freedom and the potential issues that we discussed earlier, we combined the preregistration template 'OSF Prereg'[10] with parts of the secondary data analysis template [31], and used this as a basis for our model application preregistration template. Based on our discussion of the unique modeller's degrees of freedom in model application, our template also involved adapting, removing, and adding some sections of these existing templates. We discuss the sections that we have added below, and the full template can be seen in appendix B. The sections below correspond to most of the modeller's degrees of freedom proposed above. Specifically, section A corresponds to the modeller's degrees of freedom M1, M2 and M3; section B includes questions aiming at E1, E2, E3; and section C asks about robustness checks (i.e. RC1). The modeller's degrees of freedom I1 and I2, related to statistical inference, are covered in slightly amended existing preregistration template items.

---

[10]See http://docs.google.com/document/d/1DaNmJEtBy04bq1l5OxS4JAscdZEkUGATURWwnBKLYxk/edit. This is a detailed and structured preregistration template which is based on the Prereg challenge template which was positively evaluated by Bakker *et al.* [17].

This template is explicitly an extension of the existing and widely used 'OSF Prereg' template, an approach that other field- or method-specific templates have taken [20,21,29,79]. We believe that it is sensible to take this approach, i.e. to (a) create specific templates, and (b) take existing templates as a starting point. Firstly, taking existing templates as a starting point is helpful for standardization. Second, creating specific templates based on existing general templates makes the application of preregistration more straightforward for researchers as they can build on something that other researchers in their area have used and hopefully improved on before. Third, a more specific template may encourage researchers to be more exhaustive in their preregistrations, as a general template is unlikely to include all relevant prompts. Finally, proposing specific templates based on an accepted general template may hopefully serve to make abstract discussions in a field more concrete.

It should be noted that in order for a preregistration to be exhaustive, we believe that it is important to repeatedly use clarifying words such as 'only' to explicitly constrain the choices made to *only* those mentioned in the preregistration [19].[11] A template for preregistration can be helpful in emphasizing this, as it asks specific questions and hopefully encourages exhaustiveness. Accordingly, the additions proposed here prompt answers that are as specific as possible. In particular, we added the following sections:

---

A Cognitive model (Required; *template item 29*)

A.1 Please include the type of model used (e.g. diffusion model, linear ballistic accumulator model), and a specific parameterization/parameterizations.

A.2 **Example** *As in Evans et al. [55], the parameters of a simple diffusion model will be estimated, namely only: drift rate (v), starting point (z), threshold (a), non-decision time (ter). This differs from Evans & Brown [80], where the full diffusion model was estimated, i.e. including between-trial variability parameters for drift rate, starting point, and non-decision time. These between-trial variability parameters were not relevant for Evans & Brown [80], and without them, the simple diffusion model has better parameter recovery results [81]. Figure 1 shows a plate diagram of the hierarchical structure used for the qualitative model-based analysis assessing (1) whether groups appear to get closer to optimality over time, (2) whether each group differs from optimality, and (3) whether there appears to be a difference between the groups (see Analysis Plan for more information); i indexes participants, and j indexes blocks. Only the threshold parameter varies between blocks, to estimate changes in the speed-accuracy trade-off.*

A.3 **More information** The architecture of the model should be pre-specified in a way that is specific, precise, and exhaustive (as can be seen in A.2). For example, in A.2 we emphasize that *only* the stated parameters will be estimated. To this end, you should also ideally include a plate diagram as mentioned in A.2 (e.g. figure 1) and specify the relevant equations. Motivate your choices. Note: If you are using e.g. Bayesian hierarchical modelling for parameter estimation, the structure of the hierarchical model and the prior distribution over the parameters belong in this parameterization as well.

B. Method of parameter estimation (Required; *template item 30*)

B.1 Please specify and motivate your method of parameter estimation.

B.2 **Example** *Only Bayesian hierarchical modelling will be used to estimate the parameters of the diffusion model, constraining individual-level parameters to follow group-level truncated normal distributions. For the estimation model (figure 1), the two groups (fixed-trial and fixed-time) are given a separate hierarchical structure, and the group-level parameters are not constrained between groups. Following Evans & Brown [80] and Evans et al. [55], we will use likelihood functions taken from the 'fast-dm' toolbox [82] for the calculation of the density function of the simple diffusion model. For the first model, for sampling from the posterior distributions over parameters, we will use Markov-chain Monte Carlo with differential evolution proposals [83], using 66 chains, drawing 3,000 samples from each, and discarding the first 1500 samples (as in Evans & Brown [80], see electronic supplementary material).*[12]

B.3 **More information** If you are not interested in the parameter estimates and are purely focused on statistical inference about differences between conditions/groups using a method that does not directly require initial parameter estimation, please state this clearly and motivate this choice. If you are using Bayesian methods, specify and motivate priors. In general, specify as much as

---

[11]While it can be argued that the word 'only' is implied in a preregistration, as the preregistered analyses are the only ones mentioned within the preregistration, we believe that further clarification by using the word 'only' can serve to reduce ambiguity in the preregistration.

[12]Note that while the number of samples taken in the Markov-chain Monte Carlo algorithm does not necessarily need to be specified *a priori*, as the important factor is whether the sampling algorithm converges on the posterior distribution, we believe that this level of specification is (1) knowledge that researchers experienced with specific models and specific sampling algorithms likely have, and (2) useful to help justify potential deviations from the preregistered model, such as in situations where the model does not converge within a reasonable number of samples.

possible, including for example the starting point (distribution) for estimation. If the data are going to be summarized into descriptive statistics, state which descriptive statistics will be used, and how.

C. Robustness checks and sensitivity analyses (*template item 35*)

C.1 Please specify any planned robustness checks and/or sensitivity analyses, if any. If a parameter recovery study was performed, please report its results and your conclusions here.

C.2 **Example** *The key analysis will be replicated (a) including participants/trials that were initially excluded in line with our exclusion criteria, and (b) using a model in which the threshold parameter and the drift rate parameter vary across blocks. Their results will be mentioned alongside the key results, and interpreted accordingly. If these results show a lack of robustness, this will be an interesting outcome worthy of further investigation.*

C.3 **More information** This section ensures that robustness checks and parameter recovery simulations are not performed and/or reported selectively. It is important to note that the preregistration of modelling and analyses should show that any lack of robustness is at least not due to *post hoc*, data-driven choices.

D. **Contingency Plans** (*template item 36*)

D.1 Please specify any and all contingency plans to ensure that your preregistration plan is robust to common issues. There is no need to cover every eventuality, but if possible, try and cover some of these common issues suggested here, and/or issues that you commonly encounter. Contingency plans may be easiest to specify precisely when phrased as the broader issue (e.g. whether or not the data are captured well), the more specific conditions that will be assessed to determine whether the contingency plan is used or not (e.g. the specific trends in the data that will be assessed for sufficient fit), and the contingency plan—or plans—that will be used.

D.2 **More information and some example plans**

**Issue**: The model fits poorly to the data. **Condition**: The model is unable to capture some specific trend(s) in the data that are deemed important *a priori*, such as specific response time quantiles. **Contingency plan**: A different, preregistered parameterization of the same model, or a different model altogether, which you believe will be able to account for these trends.

**Issue**: The sampling algorithm does not appear to be effectively searching through the parameter space. **Condition**: The search algorithm appears to be getting stuck in certain areas of the parameter space and gives different answers for different runs. **Contingency plan**: A different preregistered sampling algorithm, or perhaps a different preregistered parameterization of the model that may have a parameter space that is easier to search.

**Issue**: The proposed methods end up taking more time than planned. **Condition**: Executing the originally preregistered plans would take a month of computing time. **Contingency plan**: A preregistered version that is simpler, such as a reduced version of the model, or a simpler method of performing inference.

By adding these sections to a combination of the new standard, detailed OSF preregistration template and the template for secondary data use, and by adapting relevant sections to the needs of model application research, we provide a template for limiting researcher and modeller's degrees of freedom in model application. The section on cognitive models requires a choice of cognitive model and the specification of the exact parameterization(s) used, and encourages the description of specific motivations for each choice. Any *post hoc* addition or modulation is then considered to be exploratory rather than confirmatory, unless the changes can be justified as being due to technical mistakes in the original preregistration document (e.g. the researchers state that they wish to implement a specific model, but then incorrectly specify the model within the preregistration, see Cooper & Guest [62], Guest & Martin [41] for a discussion). However, in the case of technical mistakes in the original document, we believe that the best practice would be for researchers to create an updated version of the preregistration document when they realize this mistake, as well as adding any new experience that they may have with the data since the original document was written (as in 12–14 of the preregistration template). In the parameter section, the researchers are asked to provide information on the method of parameter estimation, including specific settings and/or priors used. The robustness section asks researchers to specify which robustness checks and sensitivity analyses will be performed, and how these will be interpreted. The contingency plans item prompts researchers to consider and create a contingency plan for some common issues.[13] In the fully adapted and combined preregistration template (see appendix B), researchers are further asked to specify for example how they are going to do statistical inference on which parameters.

---

[13]We wish to note that a preregistration can not (and should not) cover every eventuality. The contingency plans section should be used for specific, commonly encountered issues that have specific solutions.

# 5. Example application

## 5.1. Example application background

We include an example application of our template to showcase that using our proposed preregistration template for model application is feasible, and provide an (imperfect) example of what a preregistered model application using our template might look like. Our example application focuses on applying evidence accumulation models (also commonly referred to as sequential sampling models), which have been hugely useful and influential in the (cognitive) psychology literature [84,85], and are thus an ideal focus for our discussions of preregistration in model application. Evidence accumulation models describe the fundamental process of making a decision between alternatives in the presence of noise (e.g. [86–89]), where evidence accumulates for the different decision alternatives until the evidence for one reaches a threshold, and a decision is made. These models provide an account of the well-known speed-accuracy trade-off, where decision-makers must strike a balance between speed and accuracy, as collecting more information leads to more accurate results at the cost of more time, whereas deciding more quickly might lead to more errors. The key parameters of evidence accumulation models are the *drift rate*, which reflects how good people are at the task, the *decision threshold*, which reflects how cautious people are at the task, and the *non-decision time*, which reflects how long is spent on perceptual and motor processes.

In the example application, we aimed to replicate and analytically extend a surprising result on the speed-accuracy trade-off in Evans & Brown [80] using a previously collected dataset. Although it is well established that participants can adjust their speed-accuracy trade-off based on experimenter instructions, regardless of factors such as age ([57]; though see also [90]), a related (and contentious; [91]) question is whether participants can adjust their speed-accuracy trade-off in an *optimal* manner, in the context of maximizing their reward rate. Evans & Brown [80] used a random dot motion task with a fixed difficulty level for all decisions (in order to ensure that a single fixed boundary could produce an optimal threshold setting; [92]) to investigate whether participants can optimize their speed-accuracy trade-off in different settings. This involved experimentally varying the amount of feedback participants received and whether participants were given a fixed amount of time or a fixed number of trials in each block, and then estimating the parameters of the diffusion model using Bayesian hierarchical modelling. Participants in all conditions were instructed to try to optimize the reward rate, as they were told to maximize the number of correct responses (referred to as 'points') per minute. The main finding of Evans & Brown [80] was that with enough practice and feedback, people were able to optimize the speed-accuracy trade-off, and that they do so faster with increasing amounts of feedback. Interestingly, they also found that participants who completed a fixed number of trials per block were closer to optimality than participants who completed trials for a fixed amount of time in each block, which is in conflict with previous research (Starns & Ratcliff [58], though note that participants were given different task instructions, as discussed in more detail in the next section). This surprising result is the focus of our example application; specifically, we aim to replicate the superiority of 'fixed-trial' conditions over 'fixed-time' conditions in leading participants closer to reward rate optimality, using a different dataset and updated analysis methods.

As all feedback groups in Evans & Brown [80] showed the same general effect, the focus in this example dataset is on the effects of a 'medium' amount of feedback for fixed trials and fixed time groups, replicating the middle row of their figure 3. The assessment in Evans & Brown [80] was rather qualitative and not very rigorously defined, making this a good opportunity to show how preregistration in cognitive modelling can add rigour and transparency in situations with many potential researcher degrees of freedom. Note that we are not intending to claim that our preregistration template enabled us to provide a more rigorous or transparent assessment than any researcher(s) could have possibly managed without using our template, but rather showcase how our use of the template allowed us to add (what we believe to be) more rigour and transparency to the assessment. Furthermore, we analytically extended this replication by calculating Bayes factors to more robustly establish whether the groups differ in their distance from optimality, and if so, determine the strength of evidence in favour of the effect (which is not provided by the method in Evans & Brown [80]).

## 5.2. Example application sample and materials

We used a subset (70 participants) of an existing dataset (133 total participants) of participants who were recruited at the University of Newcastle and received course credit for their participation.

The chosen subset consists of the participants who received the same task instructions as Evans & Brown [80]; the remainder of the dataset consists of participants who were not instructed to optimize reward rate, as a manipulation of instructions. The task used was the random dot motion task (RDK; [93]) using the white noise algorithm, following Evans & Brown [80] and Evans *et al.* [55]. The participants were randomly (and equally) divided into the two groups of fixed trials and fixed time. The aim was to recruit more than 30 participants per group, which is larger than most previous studies in this area of research (e.g. [58,80]). Note that power analyses are not currently possible for the application of intricate cognitive models, and more generally, the concept of power is only applicable within a significance testing framework with meaningful cut-off points between an effect being present and not being present. Instead, we planned to use Bayes factors for a more continuous, strength of evidence approach.

As with Evans & Brown [80], participants in the included conditions were instructed to try to optimize the reward rate, as they were told to maximize the number of correct responses (referred to as 'points') per minute. Specifically, at the beginning of the task, participants in both the fixed-time and fixed-trial were told: 'Your goal is to attain as many correct answers as quickly as you can. You will receive 1 point for each correct answer, and you should try to get as many points as you can in each minute period. Remember that being too cautious may take too long and take away opportunities for points in each minute. Also, being too quick may cause you to be less accurate and take away total points.'. Between blocks, participants were also informed about their reward rate, as with the 'medium' feedback condition in Evans & Brown [80], being told: 'In recent blocks you've attained $X$ points in $Y$ minutes, meaning you achieved $X/Y$ points per minute.'. Note that these task instructions qualitatively differed from those used by Starns & Ratcliff [58], as Starns & Ratcliff [58] were interesting in comparing a condition where participants were free to choose their own speed-accuracy tradeoff (i.e. fixed-trial) to a condition where participants had an explicit performance goal (i.e. fixed-time). By contrast, Evans & Brown [80] and Evans *et al.* [55] were interested in matching the performance goals for the fixed-time and fixed-trial participants as closely as possible, to see whether previous differences between fixed-time and fixed-trial blocks were the result of the task format or the task instructions, and therefore, matched instructions as closely as possible between groups.

For all participants, the first block of trials was excluded to allow for participants to become adequately practiced at the task. Trials with response times below 150 ms or above 10 000 ms were excluded as anticipatory responses and trials where participants lost attention, respectively.[14] Participants with task accuracy below 60% (following Evans *et al.* [55]) or less than 200 eligible trials (based on the number of trials required for accurate parameter estimation) were excluded. Applying these criteria resulted in the exclusion of nine participants from the fixed-time condition (8 due to accuracy, 1 due to too few eligible trials), and 10 participants from the fixed-trial condition (all due to accuracy).

## 5.3. Example application preregistration

The example application was preregistered at https://osf.io/39t5x/ (or see appendix A for the full preregistration document). Please note that thanks to helpful reviewer comments,[15] we have added a full section on contingency plans to the general preregistration template. As this section was only added during the review process, we were not able to include it in the example application preregistration. We include parts of the preregistration here. For example, we preregistered the following hypotheses:

H1 With suitable practice and medium feedback (cf. Evans & Brown [80]), participants get closer to optimality with each block of trials. (directional)
H2 After suitable practice and medium feedback (cf. Evans & Brown [80]), participants will have an approximately optimal speed-accuracy trade-off. (directional)
H3 Participants who complete a fixed number of trials are closer to optimality than participants who complete trials in a fixed amount of time. (directional)

To test these hypotheses, we preregistered the following:

Testing H1/2/3: In addition to the statistical analyses, we will qualitatively compare the posterior distributions of the decision threshold parameters (actual thresholds for each block as estimated

---

[14]Note that this is a particularly important aspect to preregister in model application, as a wide variety of 'rules' for RT outlier exclusion can be found in the literature, and even within a single paper.

[15]Specifically, we thank Jeffrey Starns for suggesting the inclusion of this section.

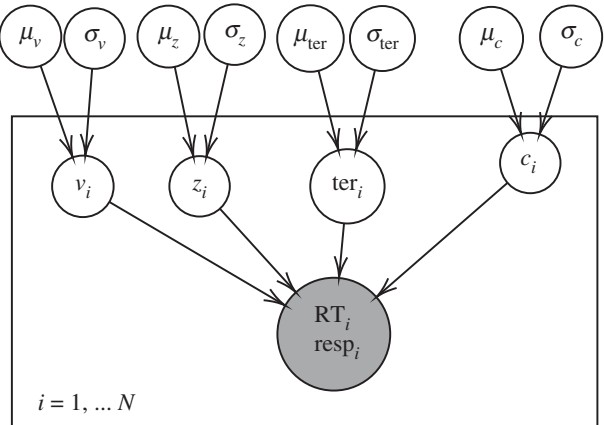

**Figure 2.** Plate diagram of the hierarchical structure used for the quantitative model-based analysis to test whether each group differs from optimality using Bayes factors, approximated with the Savage–Dickey Ratio on $\mu_c$. RT stands for reaction time, resp stands for response accuracy and $i$ indexes participants.

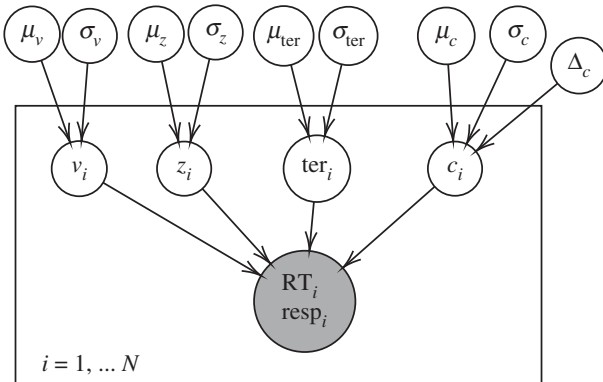

**Figure 3.** Plate diagram of the hierarchical structure used for the quantitative model-based analysis to test the hypothesis whether the groups differ from each other in their difference from optimality. RT stands for reaction time, resp stands for response accuracy and $i$ indexes participants.

using the model in figure 1) against the posterior predictive distributions for the optimal threshold calculated as above.

Testing H2: Using only the second half of all 20 blocks (11–20, so as to account for participants adjusting to the task), we will test whether each group, separately, differs from optimality using Bayes factors, approximated with the Savage–Dickey Ratio on $\mu_c$ (Wagenmakers *et al.* [94], see figure 2 for the corresponding plate diagram).

Testing H3: Again using only blocks 11–20, we will use the Savage–Dickey method on $\Delta_c$ ($\Delta_c = \mu_{c_1} - \mu_{c_2}$) to test whether the groups differ in their distance from optimality.

The preregistered models can be seen in figures 1–3, and further preregistered information such as on the distributions of the parameters and the relations between them, and on the parameter estimation method can be found in the full preregistration document (see appendix A).

## 5.4. Example application results

Figure 4 shows the posterior distributions of the decision threshold parameters against the posterior predictive distributions for the optimal threshold. From our first analysis, qualitatively testing H1/2/3, we can qualitatively observe the same trend to optimality over time as in Evans & Brown [80] and Evans *et al.* [55]. From this qualitative analysis, however, it is unclear whether the groups differ in the extent to which they move towards optimality. Our second analysis, quantitatively testing H2 by looking at each group separately, reveals strong evidence for the participants of both groups being too cautious in their decision making ($BF_{Time} = 149$, $BF_{Trial} = 31.664$). Our third analysis, quantitatively testing H3 by using the Savage–Dickey method on $\Delta_c$ leads to weak evidence for the groups not differing in their distance from

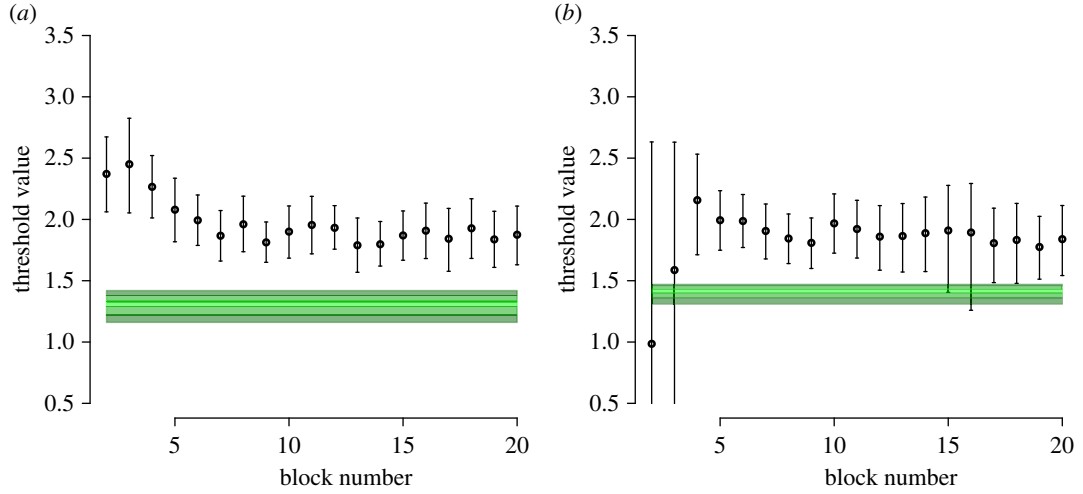

**Figure 4.** Plots comparing the posterior distributions of the decision threshold parameters (dots with error bars, reflecting the posterior median and the 95% credible interval, respectively) against the posterior predictive distributions for the optimal threshold (green area; lightest shade being the 0.4–0.6 quantile region, the middle shade being the 0.2–0.4 and 0.6–0.8 quantile regions, and the darkest shade being the tails of the distribution), for the fixed time group (a) and the fixed trial group (b).

optimality (BF = 1.182), albeit in the direction of the fixed trial group being closer to optimality than the fixed time group. Fixed-time participants completed an average of 25.66 trials per block, and fixed-trial participants took an average of 92.93 s per block.

## 5.5. Example application discussion

Our results do not fully replicate Evans & Brown [80]. In our replication, participants did move towards optimality given practice and feedback, but there was no clear difference between the fixed-time and fixed-trial groups. This may be a result of our updated statistical analysis methods, as our qualitative pattern of results look similar to those from Evans & Brown [80], and it is our new quantitative analyses that suggest that there is no evidence for a difference between groups. Regardless, our findings indicate that there is not necessarily a difference between fixed-trial blocks and fixed-time blocks in how close people are able to come to reward rate optimality, and so this perceived difference in Evans & Brown [80] should be interpreted with caution. It should also be noted that our findings showed strong evidence for participants being suboptimally cautious, which again is somewhat against the conclusions of Evans & Brown [80], but makes sense in the context of Evans & Brown [55] who suggested that only specific experimental designs (e.g. slower trial-to-trial timing) will result in people achieving reward rate optimality.

# 6. General discussion

In this article, we have proposed a concrete list of modeller's degrees of freedom, developed a preregistration template for model application, and showcased the possibility of preregistering model application studies with an example application. Our overarching goal was to display how preregistration templates can be developed in areas of psychology that have diverse hypotheses and diverse analyses, such as cognitive modelling, with a more specific goal of making preregistration in model application more feasible. We also believe that our concrete list of modeller's degrees of freedom may be a useful resource for researchers when they consider which aspects of a model application study may be prone to flexibility, regardless of whether or not they choose to perform a preregistration. Nevertheless, as we argued previously, we believe that preregistration is the best tool currently available for attempting to constrain researcher degrees of freedom, and we believe that model application studies may benefit from the use of preregistration and our template. Importantly, we believe that our article provides a first step towards bridging the gap between open science and

mathematical psychology, and we hope that our work will provide new insights to the debate on whether preregistration has utility within cognitive modelling research.

Nevertheless, deviations from a preregistration should always be possible, for example if the researcher gained important knowledge in between writing the preregistration and analysing the data, or if the data violate assumptions of the planned analyses—'the preregistration of a model and the way it will be used may not survive contact with the data' ([24], p. 4). Moreover, minor accidental omissions which likely do not affect the outcome can happen with any preregistration, and should be transparently reported (for a discussion and a few examples of how to transparently report and justify deviations, see [95]). Although we believe that our preregistration template allowed us to create a highly constrained preregistration document for our example application, we wish to note that the amount of researcher degrees of freedom available—and therefore, the amount of detail required in the preregistration—was at times surprising. Even after carefully going over the example application preregistration several times, we still accidentally omitted a minor detail about the process of parameter estimation: that we used a migration algorithm to assist convergence (as did [55,80]). We hope that the clear imperfection in our own example application demonstrates several important aspect of preregistration: that (i) a preregistration is not a prison, (ii) it is always possible to make changes given that they are well motivated and reported in a transparent manner ([96]; though see [95] for a critical perspective on the lack of transparent reporting of deviations from preregistered plans in practice), and (iii) creating preregistration documents is a learning process that improves with time. Similarly, exploratory work is important and should not suffer as a result of an increased move towards preregistered confirmatory work, and we believe that the interplay of unrestricted creativity and constrained verification remains important [97].

One potential critique of our preregistration template could be that the prompts are too open-ended. However, while the questions are open-ended, they are asked in a structured format. It has previously been found that a structured preregistration template format with specific but open-ended questions is better at restricting researcher degrees of freedom than an unstructured, open-ended template [17]. Supporting this, it has been argued that some increase in transparency and constraint is better than none [98]. What is more, templates that are too long or ask too many questions that are too specific might deter researchers from using the template. Therefore, we believe that our preregistration template strikes a sensible balance between a completely open-ended template and an overly long template, as our template prompts researchers to disclose key information that relates to the modeller's degrees of freedom that we discuss within our article, but does not ask researchers hundreds of small questions regarding every potential detail of their study.

## 6.1. Limitations

First and foremost, it should be noted that our proposed preregistration template is an initial proposal of what a preregistration might look like in cognitive modelling, and specifically for the category of model application. We do not intend for our template to be a definitive answer to preregistration in cognitive modelling, or even in model application. Instead, our aim is to create an initial tool for researchers who are interested in preregistering their model application study, but are unsure of how to do so. Furthermore, although we attempted to make our template as generalizable as possible, it should be noted that many of our modeller's degrees of freedom were inspired by our previous experience working with evidence accumulation models—the class of models that we applied within our example application. Researchers more familiar with other classes of models may have different opinions on which degrees of freedom should be constrained within a model application preregistration document. It is also possible that researchers may experience unique issues that are not covered by our preregistration template when attempting to preregister studies that differ greatly from our example application, such as when using a different class of model. Therefore, we hope that others will critique and build upon our initial work, leading to the further development of a range of preregistration templates for cognitive modelling. Moreover, we believe that future research should aim to empirically evaluate both our proposed preregistration template and modeller's degrees of freedom, for example by comparing preregistered to non-preregistered cognitive modelling studies. This may also highlight further limitations of our work that could be overcome in future extensions.

A second limitation of our preregistration template is that it only covers the cognitive modelling category of model application. As stated earlier, we believe that model application is an ideal category for initial preregistration efforts within cognitive modelling, as it is highly confirmatory in nature and is the most closely related to purely experimental research, making it most closely conform to the focus of most previous preregistration efforts. But cognitive modelling studies often involve

the interplay between several categories of cognitive modelling, which may lead some to question the utility of a preregistration template for a single category of cognitive modelling. For example, a researcher might want to assess whether a certain parameter estimated from the cognitive model differs over experimental conditions, but only if the model provides an adequate account of the data that the parameters are being estimated from, which would constitute an interplay of model application with model evaluation. We agree that the potential interplay between different categories of cognitive modelling limits the possible scope of our preregistration template. As discussed earlier, however, we believe that our template is only an initial step towards preregistration in cognitive modelling, and that future research may be able to find solutions for the potential interplay between categories. In the above example, the researcher could combine our preregistration template for model application with a preregistration template for model evaluation. This would allow the researcher to evaluate the fit of the model in a constrained way based on *a priori* defined criteria. If the fit is found to be inadequate, the researcher is able to make alterations to the model based on *a priori* defined rules, thereby preventing any unnecessary deviations from the preregistration for the model application. Although the above scenario is not currently possible, as there is no preregistration template for model evaluation (though see [99] for an initial proposal of some factors that might be important to consider for preregistering model evaluation) or guidelines for how to combine these templates, we believe that (i) a preregistration of just the model application aspect of the study would give us more information than no preregistration at all, and (ii) many cognitive models used in model application are already sufficiently developed to provide an accurate account of a range of data, meaning that the model evaluation step may not always be necessary.

## 6.2. Future directions

When considering the future of preregistration in cognitive modelling, we believe that mathematical psychology is one of the fields of psychology best suited to creating constrained preregistration documents. Cognitive modelling is a highly theory-driven field of research [100], and the formal nature of cognitive models means that they make precise predictions about empirical data. Comparing this to the widespread lack of theory in other parts of psychology [101] suggests that certain categories of cognitive modelling—model application, model comparison, and model evaluation—may lend themselves particularly well to preregistration. However, we believe that this will only be the case when appropriate tools, such as preregistration templates, are available and tailored to cognitive modelling research. Therefore, we believe that future research efforts should focus on developing preregistration templates for the other categories of cognitive modelling proposed by Crüwell *et al.* [10], either through extending our preregistration template for model application or by creating new preregistration templates. With specific templates for each category, a project including more than one modelling category could use the templates for each category to constrain the degrees of freedom in each part of the process, either simultaneously or sequentially. For example, a researcher might preregister a model comparison between a series of candidate models to decide which best accounts for the data, and after conducting this comparison might preregister a model evaluation for each of the models on a series of data trends to determine why the best model performed better than the other models. We believe that this system of iterative preregistrations for different categories of cognitive modelling within a single study provides the ideal balance between constraint and diversity, as researchers are free to investigate the data in as much detail as they wish while each analysis performed is constrained.

At a general level, we believe that there should be an increased focus on the development of preregistration templates for specific fields and/or methods of research. Although excessively general templates might seem appealing, as they appear to unite different fields of research within a single framework, these general templates will often lack the specificity needed to properly constrain the researcher degrees of freedom for each field. No preregistration is perfect, and using an overly general template—or one designed for a different field of research—may help to provide more information than using no template or eschewing preregistration entirely. Nevertheless, we believe that future preregistration efforts should focus on developing specific preregistration templates for more fields of research, so that researchers can avoid unnecessarily decreasing their ability to effectively constrain researcher degrees of freedom.

Data accessibility. We report how we determined our sample size, all data exclusions, all manipulations and all measures in the example application study. The preregistration document, data and code are available on the OSF: https://osf.io/39t5x/. The preregistration example application and preregistration template for model application have been uploaded to FigShare [102].

Authors' contributions. S.C. led the theoretical aspects of the investigation (including the development of the preregistration template), provided the project administration, and wrote the original draft of the manuscript. N.J.E. provided supervision for the project and led the analysis of the example application, writing the software and performing the data curation and formal analysis. Both authors conceptualized the study, obtained necessary resources, provided visualization of the models and data, and reviewed and edited the manuscript.

Competing interests. We declare we have no competing interests.

Funding. N.J.E. was supported by an Australian Research Council Discovery Early Career Researcher award no. (DE200101130).

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
