## [Peer Review File · Royal Society Open Science]

Review History

RSOS-210155.R0 (Original submission)

Review form: Reviewer 1 (Alex Holcombe)

Is the manuscript scientifically sound in its present form?

Yes

Are the interpretations and conclusions justified by the results?

Yes

Is the language acceptable?

Yes

Do you have any ethical concerns with this paper?

Yes

Have you any concerns about statistical analyses in this paper?

No

Recommendation?

Accept with minor revision (please list in comments)

Comments to the Author(s)

See attached file (Appendix A).

Review form: Reviewer 2 (Olivia Guest)

Is the manuscript scientifically sound in its present form?

No

Are the interpretations and conclusions justified by the results?

No

Is the language acceptable?

Yes

Do you have any ethical concerns with this paper?

No

Have you any concerns about statistical analyses in this paper?

No

Recommendation?

Reject

Comments to the Author(s)

I have attached it as a separate file (see Appendix B).

Review form: Reviewer 3

Is the manuscript scientifically sound in its present form?

Yes

Are the interpretations and conclusions justified by the results?

Yes

Is the language acceptable?

Yes

Do you have any ethical concerns with this paper?

No

Have you any concerns about statistical analyses in this paper?

No

Recommendation?

Major revision is needed (please make suggestions in comments)

Comments to the Author(s)

This is an excellent paper about preregistration cognitive model application. The authors discuss the specific researcher degrees of freedom for this type of research and make a clear case why we need this additional preregistration template. They are also very clear for which type of cognitive models it is and isn't a good template.

On pages 11 and 12, the additional degrees of freedom are mentioned, and on pages 19 and following, the new sections for the preregistration template. I got a bit confused by the different letters that are used (M, E, I, and RC for the RDFs, and A, B, C, and D for the sections). They seem to overlap partly, but not completely. I think that it will be helpful to relate the new RDFs to the specific new sections. In this way, it is clearer how these sections will prevent the opportunistic use of these additional RDFs.

I wasn't sure of the function of the example application (starting at page 24). On page 6, the authors mention that it is to "showcase how it can help to constrain researcher degrees of freedom". Therefore, I expected an example preregistration and a discussion of the RDFs that are prevented. But it is more a general research example (including results and discussion) with only limited information about the preregistration itself. The preregistration itself is put in the appendix. It would be helpful if the authors clarify why they included an example application (the goals) and add more of the preregistration in this example preregistration.

Additional comments:

- In the second paragraph of page 3, preregistration is defined and explained. I miss that preregistrations should be time-stamped/frozen registrations.
- In the same paragraph, the different reasons to preregister are mentioned. However, I miss the transparency reason, although transparency is mentioned as an important reason for preregistration in the remainder of the paper (e.g., p. 8).
- The study by Veldkamp et al. is now published: Bakker, M., Veldkamp, C. L., van Assen, M. A., Crompvoets, E. A., Ong, H. H., Nosek, B. A., ... & Wicherts, J. M. (2020). Ensuring the quality and specificity of preregistrations. *PLoS Biology*, 18(12), e3000937. <https://doi.org/10.1371/journal.pbio.3000937>
- On page 4, row 12, I would add the PNAS paper by Nosek et al. because they use postdiction and prediction in that paper.
- On page 14, secondary data is discussed. An essential aspect of preregistering secondary data is discussing prior knowledge of the data (this is also part of the model application template). This helps authors to be transparent about what they know and do not know of the data set and allows readers to evaluate this. I think that it is good to mention this aspect of registering secondary data as well.
- On page 32, changes after preregistration are discussed, and it is stated that these should be motivated and reported in a transparent manner (row 24). Can you give some examples or references on how to report this transparently? I think that this is very helpful because this is something that currently often goes wrong, as you discuss here as well.

Review form: Reviewer 4 (Wolf Vanpaemel)

Is the manuscript scientifically sound in its present form?

Yes

Are the interpretations and conclusions justified by the results?

Yes

Is the language acceptable?

Yes

Do you have any ethical concerns with this paper?

No

Have you any concerns about statistical analyses in this paper?

No

Recommendation?

Major revision is needed (please make suggestions in comments)

Comments to the Author(s)

The authors provide a preregistration template for situations in which data analyses require the application of a cognitive model rather than an off-the-shelf model like e.g. a regression model.

Quite bluntly, I am not convinced of the usefulness of a specific template for this situation. My hesitation does not stem from the fact that I don't believe in the usefulness of preregistration when engaging in cognitive modeling (in fact, I have proposed a format for a Registered Report specifically geared towards theory testing, designed to assure severe tests of cognitive models; see Vanpaemel, 2019). Rather, I am skeptical because the current template overlaps quite extensively with the standard OSF checklist (<https://docs.google.com/document/d/1DaNmJEtBy04bq115OxS4JAscDZEKUGATURWwnBKL Yxk/edit?pli=1#>). I realize this is very much my personal opinion, but I doubt the few deviations warrant a checklist of its own.

As far as preregistration is concerned, I don't see a fundamental difference between registering the use of, say, a SEM type model and a cognitive model. Both require the exact specification of the model version (including which parameters to include) and the estimation technique (say OLS vs multilevel approach). This intuition is further strengthened by the fact that the distinction between cognitive and statistical models is sometimes moot: MDS started as a cognitive one, but is now treated as a statistical one; the cognitive model FLMP turned out to be equivalent to the statistical Rasch model, and so on. That, together, with a desire for parsimony, makes me doubt the usefulness of this template.

Personally, I would much rather read a very brief paper with some caveats and good practices when using an existing pre-reg template when a cognitive model is being used, rather than having yet another template to choose from. Nevertheless, I am not to decide on the usefulness, so I will provide some comments for improvement, assuming others find the new template more useful than I do.

It is unlikely that this version of the template is the ideal or the final one (or even if it is ideal, will stay ideal in a continuously moving landscape). Therefore, I would encourage the authors to actively seek feedback from users of the template, and adapt the template continuously based on the feedback (with a proper versioning system).

I haven't read the papers introducing the four types of cognitive modeling, but I was wondering why this example wasn't a case of model comparison (clearly, using the Bayes factor, two models are being compared, so surely I am missing something).

Not everybody agrees that the distinction between exploratory and confirmatory analyses is meaningful and/or useful. Maybe this position should be acknowledged or discussed (see Szollosi and Donkin, 2019).

I am not sure whether the authors simply borrow terminology from Veldkamp, or speak with their own voice, but I would be very hesitant to use the term "the quality of preregistrations". Veldkamp et al. have looked at scope and level of detail, which are (fairly) objective characteristics, but equating these with quality is a strong epistemological/meta-scientific commitment.

I think "E2: Specifying settings/priors for parameter estimation." is misguided. As I have extensively argued elsewhere, especially in cognitive modeling, priors should be seen as part of the model (in some sense, models *are* predictions, and without priors, we can not make predictions; Vanpaemel, 2010; Vanpaemel and Lee, 2012). This is not to say that priors should not be part of the preregistration plan, but I think they are more fit in the M category. (The same holds for B.3) The authors are of course free to disagree with me on the role of the prior in cognitive modeling, but I would like to make sure that putting priors in E and not in M is a deliberate decision on their part.

I think it is a dangerous practice to interpret or do tests on parameters estimated using models that might not be appropriate for the data at hand. It seems that at least a minimal check of descriptive adequacy (using e.g., posterior predictive checks) should be included, or more broadly, the conditions under which one is confident enough in the model to work with the estimates. (To be fair, this comment is not restricted to cognitive modeling, and descriptive adequacy should also be checked when e.g regression is used.)

I (for one, but that is again just my personal opinion) strongly disagree with the necessity of words such as "only" in pre-reg plans. It makes pre-reg plans overly legalistic, and, most importantly, these words seem to be implied, by Gricean maxims. If my son asks me what he can choose for dessert, and I would respond "There is cake", he would be surprised (and angry/or annoyed with my dad joke) when he would find out there is not just cake but pie as well, even though what I said is not a lie and technically correct. He is just working on the assumption that if I answer his question, I am being exhaustive (unless I use words like "for example"). I believe something similar happens when someone writes in a pre-reg plan "we will vary parameter theta". If these authors then end up varying both theta and delta in their paper, this is technically correct (i.e, consistent with the pre-reg plan), but I think most readers will be surprised to see this, as they will work on the assumption that "we will vary parameter theta" is exhaustive.

What exactly is meant with "parameterisation of the model"? My interpretation would be to the version used to describe the *same* model (e.g., a Beta distribution can be described using alpha and beta, but also using a mode and a concentration; see https://en.wikipedia.org/wiki/Beta_distribution#Alternative_parametrizations), but I think the authors mean to refer to which parameters are included to make up *different* models (e.g., some versions of the GCM use a response scaling parameter whereas others don't). I think it is important to avoid this potential confusion.

How can the appropriate number of samples be meaningfully set before the chain is run? At least partially, the number of samples seems to depend on data-dependent issues, such as convergence?

Further, I think the preregistration included as an example is fine as is. However, to serve as an *exemplary* preregistration, fine maybe not be good enough. A few examples of where I think it could be improved: I realize not much can be done, given that it is preregistration, but the absence of a robustness analysis makes the example application a rather poor example. The "Example Application Results" section should do a better job linking the reported BF's to the

different hypotheses. The redundancy between 1.4 and 8.1 is confusing. Why is the test of H1 classified as "other analyses"?

Signed,
wolf vanpaemel

References

Szollosi, A., & Donkin, C. (2019, September 21). Arrested theory development: The misguided distinction between exploratory and confirmatory research. <https://doi.org/10.1177/1745691620966796>

Vanpaemel, W. (2010). Prior sensitivity in theory testing: An apologia for the Bayes factor. *Journal of Mathematical Psychology*, 54, 491-498. <https://doi.org/10.1016/j.jmp.2010.07.003>

Vanpaemel, W. (2019) The Really Risky Registered Modeling Report: Incentivizing Strong Tests and HONEST Modeling in Cognitive Science. *Comput Brain Behav* 2, 218–222. <https://doi.org/10.1007/s42113-019-00056-9>

Vanpaemel, W., & Lee, M. D. (2012). Using priors to formalize theory: Optimal attention and the generalized context model. *Psychonomic Bulletin & Review*, 19, 1047-1056. <https://doi.org/10.3758/s13423-012-0300-4>

Decision letter (RSOS-210155.R0)

Dear Ms Crüwell

The Editors assigned to your paper RSOS-210155 "Preregistration in Complex Contexts: A Preregistration Template for the Application of Cognitive Models" have now received comments from reviewers and would like you to revise the paper in accordance with the reviewer comments and any comments from the Editors. Please note this decision does not guarantee eventual acceptance.

Please submit your revised manuscript and required files (see below) no later than 21 days from today's (ie 13-Apr-2021) date. Note: the ScholarOne system will 'lock' if submission of the revision is attempted 21 or more days after the deadline. If you do not think you will be able to meet this deadline please contact the editorial office to arrange an extension.

on behalf of Professor Chris Chambers (Subject Editor)
openscience@royalsociety.org

Associate Editor Comments to Author (Professor Chris Chambers):

Comments to the Author:

Four expert reviewers have now assessed the manuscript. Their opinions vary: Reviewers 1 and 3 are largely positive (yet still noting areas requiring significant revision/consideration), while Reviewers 2 and 4 are more critical, questioning the necessity (and indeed general value) of a preregistration template for cognitive modeling and also noting a wide range of areas needing greater clarity, precision and details of underlying arguments and assumptions. In my own reading of the manuscript and the reviews, I found myself returning frequently to the sentiment expressed in Reviewer 2's opening comment: "who is the audience for this paper". I can envisage who the audience might be, but I agree with Reviewer 2 (and 4) that significant work is needed to make this clearer in the presentation. It is imperative, too, that the deep concern about redundancy of the preregistration template (expressed by Reviewer 4) is comprehensively settled.

Given this set of reviews, I think many editors would be inclined to reject the manuscript; however, given the detailed and constructive nature of the reviews, I think that would be a wasted opportunity in this case. These evaluations provide an ideal chance to improve the manuscript and maximise its positive impact with the intended audience.

Reviewer comments to Author:

Reviewer: 1

Comments to the Author(s)

See attached file

Reviewer: 2

Comments to the Author(s)

I have attached it as a separate file.

Reviewer: 3

Comments to the Author(s)

This is an excellent paper about preregistration cognitive model application. The authors discuss the specific researcher degrees of freedom for this type of research and make a clear case why we need this additional preregistration template. They are also very clear for which type of cognitive models it is and isn't a good template.

On pages 11 and 12, the additional degrees of freedom are mentioned, and on pages 19 and following, the new sections for the preregistration template. I got a bit confused by the different letters that are used (M, E, I, and RC for the RDFs, and A, B, C, and D for the sections). They seem to overlap partly, but not completely. I think that it will be helpful to relate the new RDFs to the specific new sections. In this way, it is clearer how these sections will prevent the opportunistic use of these additional RDFs.

I wasn't sure of the function of the example application (starting at page 24). On page 6, the authors mention that it is to "showcase how it can help to constrain researcher degrees of freedom". Therefore, I expected an example preregistration and a discussion of the RDFs that are prevented. But it is more a general research example (including results and discussion) with only limited information about the preregistration itself. The preregistration itself is put in the appendix. It would be helpful if the authors clarify why they included an example application (the goals) and add more of the preregistration in this example preregistration.

Additional comments:

- In the second paragraph of page 3, preregistration is defined and explained. I miss that preregistrations should be time-stamped/frozen registrations.
- In the same paragraph, the different reasons to preregister are mentioned. However, I miss the transparency reason, although transparency is mentioned as an important reason for preregistration in the remainder of the paper (e.g., p. 8).
- The study by Veldkamp et al. is now published: Bakker, M., Veldkamp, C. L., van Assen, M. A., Cromptvoets, E. A., Ong, H. H., Nosek, B. A., ... & Wicherts, J. M. (2020). Ensuring the quality and specificity of preregistrations. *PLoS Biology*, 18(12), e3000937.
<https://doi.org/10.1371/journal.pbio.3000937>
- On page 4, row 12, I would add the PNAS paper by Nosek et al. because they use postdiction and prediction in that paper.
- On page 14, secondary data is discussed. An essential aspect of preregistering secondary data is discussing prior knowledge of the data (this is also part of the model application template). This helps authors to be transparent about what they know and do not know of the data set and allows readers to evaluate this. I think that it is good to mention this aspect of registering secondary data as well.
- On page 32, changes after preregistration are discussed, and it is stated that these should be motivated and reported in a transparent manner (row 24). Can you give some examples or references on how to report this transparently? I think that this is very helpful because this is something that currently often goes wrong, as you discuss here as well.

Reviewer: 4

Comments to the Author(s)

The authors provide a preregistration template for situations in which data analyses require the application of a cognitive model rather than an off-the-shelf model like e.g. a regression model.

Quite bluntly, I am not convinced of the usefulness of a specific template for this situation. My hesitation does not stem from the fact that I don't believe in the usefulness of preregistration when engaging in cognitive modeling (in fact, I have proposed a format for a Registered Report specifically geared towards theory testing, designed to assure severe tests of cognitive models; see Vanpaemel, 2019). Rather, I am skeptical because the current template overlaps quite extensively with the standard OSF checklist (<https://docs.google.com/document/d/1DaNmJEtBy04bq1l5OxS4JAscdZEKUGATURWwnBKL Yxk/edit?pli=1#>). I realize this is very much my personal opinion, but I doubt the few deviations warrant a checklist of its own.

As far as preregistration is concerned, I don't see a fundamental difference between registering the use of, say, a SEM type model and a cognitive model. Both require the exact specification of the model version (including which parameters to include) and the estimation technique (say OLS vs multilevel approach). This intuition is further strengthened by the fact that the distinction between cognitive and statistical models is sometimes moot: MDS started as a cognitive one, but is now treated as a statistical one; the cognitive model FLMP turned out to be equivalent to the statistical Rasch model, and so on. That, together, with a desire for parsimony, makes me doubt the usefulness of this template.

Personally, I would much rather read a very brief paper with some caveats and good practices when using an existing pre-reg template when a cognitive model is being used, rather than having yet another template to choose from. Nevertheless, I am not to decide on the usefulness, so I will provide some comments for improvement, assuming others find the new template more useful than I do.

It is unlikely that this version of the template is the ideal or the final one (or even if it is ideal, will stay ideal in a continuously moving landscape). Therefore, I would encourage the authors to actively seek feedback from users of the template, and adapt the template continuously based on the feedback (with a proper versioning system).

I haven't read the papers introducing the four types of cognitive modeling, but I was wondering why this example wasn't a case of model comparison (clearly, using the Bayes factor, two models are being compared, so surely I am missing something).

Not everybody agrees that the distinction between exploratory and confirmatory analyses is meaningful and/or useful. Maybe this position should be acknowledged or discussed (see Szollosi and Donkin, 2019).

I am not sure whether the authors simply borrow terminology from Veldkamp, or speak with their own voice, but I would be very hesitant to use the term "the quality of preregistrations". Veldkamp et al. have looked at scope and level of detail, which are (fairly) objective characteristics, but equating these with quality is a strong epistemological/meta-scientific commitment.

I think "E2: Specifying settings/priors for parameter estimation." is misguided. As I have extensively argued elsewhere, especially in cognitive modeling, priors should be seen as part of the model (in some sense, models *are* predictions, and without priors, we can not make predictions; Vanpaemel, 2010; Vanpaemel and Lee, 2012). This is not to say that priors should not be part of the preregistration plan, but I think they are more fit in the M category. (The same holds for B.3) The authors are of course free to disagree with me on the role of the prior in cognitive modeling, but I would like to make sure that putting priors in E and not in M is a deliberate decision on their part.

I think it is a dangerous practice to interpret or do tests on parameters estimated using models that might not be appropriate for the data at hand. It seems that at least a minimal check of descriptive adequacy (using e.g., posterior predictive checks) should be included, or more broadly, the conditions under which one is confident enough in the model to work with the estimates. (To be fair, this comment is not restricted to cognitive modeling, and descriptive adequacy should also be checked when e.g regression is used.)

I (for one, but that is again just my personal opinion) strongly disagree with the necessity of words such as "only" in pre-reg plans. It makes pre-reg plans overly legalistic, and, most

importantly, these words seem to be implied, by Gricean maxims. If my son asks me what he can choose for dessert, and I would respond "There is cake", he would be surprised (and angry/or annoyed with my dad joke) when he would find out there is not just cake but pie as well, even though what I said is not a lie and technically correct. He is just working on the assumption that if I answer his question, I am being exhaustive (unless I use words like "for example"). I believe something similar happens when someone writes in a pre-reg plan "we will vary parameter theta". If these authors then end up varying both theta and delta in their paper, this is technically correct (i.e, consistent with the pre-reg plan), but I think most readers will be surprised to see this, as they will work on the assumption that "we will vary parameter theta" is exhaustive.

What exactly is meant with "parameterisation of the model"? My interpretation would be to the version used to describe the *same* model (e.g., a Beta distribution can be described using alpha and beta, but also using a mode and a concentration; see https://en.wikipedia.org/wiki/Beta_distribution#Alternative_parametrizations), but I think the authors mean to refer to which parameters are included to make up *different* models (e.g., some versions of the GCM use a response scaling parameter whereas others don't). I think it is important to avoid this potential confusion.

How can the appropriate number of samples be meaningfully set before the chain is run? At least partially, the number of samples seems to depend on data-dependent issues, such as convergence?

Further, I think the preregistration included as an example is fine as is. However, to serve as an *exemplary* preregistration, fine maybe not be good enough. A few examples of where I think it could be improved: I realize not much can be done, given that it is preregistration, but the absence of a robustness analysis makes the example application a rather poor example. The "Example Application Results" section should do a better job linking the reported BFs to the different hypotheses. The redundancy between 1.4 and 8.1 is confusing. Why is the test of H1 classified as "other analyses"?

Signed,
wolf vanpaemel

References

Szollosi, A., & Donkin, C. (2019, September 21). Arrested theory development: The misguided distinction between exploratory and confirmatory research. <https://doi.org/10.1177/1745691620966796>

Vanpaemel, W. (2010). Prior sensitivity in theory testing: An apologia for the Bayes factor. *Journal of Mathematical Psychology*, 54, 491-498. <https://doi.org/10.1016/j.jmp.2010.07.003>

Vanpaemel, W. (2019) The Really Risky Registered Modeling Report: Incentivizing Strong Tests and HONEST Modeling in Cognitive Science. *Comput Brain Behav* 2, 218–222. <https://doi.org/10.1007/s42113-019-00056-9>

Vanpaemel, W., & Lee, M. D. (2012). Using priors to formalize theory: Optimal attention and the generalized context model. *Psychonomic Bulletin & Review*, 19, 1047-1056. <https://doi.org/10.3758/s13423-012-0300-4>

===PREPARING YOUR MANUSCRIPT===

===PREPARING YOUR REVISION IN SCHOLARONE===

- An editable file of each table (.doc, .docx, .xls, .xlsx, or .csv).
- An editable file of all figure and table captions.

- Any electronic supplementary material (ESM).
- If you are requesting a discretionary waiver for the article processing charge, the waiver form must be included at this step.
- If you are providing image files for potential cover images, please upload these at this step, and inform the editorial office you have done so. You must hold the copyright to any image provided.
- A copy of your point-by-point response to referees and Editors. This will expedite the preparation of your proof.

- Ensure that your data access statement meets the requirements at <https://royalsociety.org/journals/authors/author-guidelines/#data>. You should ensure that you cite the dataset in your reference list. If you have deposited data etc in the Dryad repository, please include both the 'For publication' link and 'For review' link at this stage.
- If you are requesting an article processing charge waiver, you must select the relevant waiver option (if requesting a discretionary waiver, the form should have been uploaded at Step 3 'File upload' above).
- If you have uploaded ESM files, please ensure you follow the guidance at <https://royalsociety.org/journals/authors/author-guidelines/#supplementary-material> to include a suitable title and informative caption. An example of appropriate titling and captioning may be found at [https://figshare.com/articles/Table_S2_from_Is_there_a_trade-off_between_peak_performance_and_performance_breadth_across_temperatures_for_aerobic_sc](https://figshare.com/articles/Table_S2_from_Is_there_a_trade-off_between_peak_performance_and_performance_breadth_across_temperatures_for_aerobic_scope_in_teleost_fishes_/3843624) ope_in_teleost_fishes_/3843624.

Author's Response to Decision Letter for (RSOS-210155.R0)

See Appendix C.

RSOS-210155.R1 (Revision)

Review form: Reviewer 1 (Alex Holcombe)

Is the manuscript scientifically sound in its present form?

Yes

Are the interpretations and conclusions justified by the results?

Yes

Is the language acceptable?

Yes

Do you have any ethical concerns with this paper?

No

Have you any concerns about statistical analyses in this paper?

No

Recommendation?

Accept as is

Comments to the Author(s)

I appreciate the authors' revision and careful consideration of my and the other referees' points, and as the authors stated in their response, this concrete initiative in the domain of cognitive modeling may do more to advance the debate regarding preregistration in that literature than would another more abstract or theoretical paper.

About the sentence on page 4 that ends with "areas of research", it is missing a period at the end of the sentence.

There is an extra closed parenthesis on page 21 where Steegen et al. are cited.

Review form: Reviewer 2 (Olivia Guest)

Is the manuscript scientifically sound in its present form?

Yes

Are the interpretations and conclusions justified by the results?

Yes

Is the language acceptable?

Yes

Do you have any ethical concerns with this paper?

No

Have you any concerns about statistical analyses in this paper?

No

Recommendation?

Accept as is

Comments to the Author(s)

The changes made by the authors greatly improve the clarity and scholarship of the piece. I have no more feedback to give on this basis and believe this is probably in its final state and publishable.

Review form: Reviewer 3

Is the manuscript scientifically sound in its present form?

Yes

Are the interpretations and conclusions justified by the results?

Yes

Is the language acceptable?

Yes

Do you have any ethical concerns with this paper?

No

Have you any concerns about statistical analyses in this paper?

No

Recommendation?

Accept as is

Comments to the Author(s)

All my comments are well addressed in the new version

Decision letter (RSOS-210155.R1)

Dear Ms Crüwell,

It is a pleasure to accept your manuscript entitled "Preregistration in Diverse Contexts: A Preregistration Template for the Application of Cognitive Models" in its current form for publication in Royal Society Open Science. The comments of the reviewer(s) who reviewed your manuscript are included at the foot of this letter.

Please ensure that you send to the editorial office an editable version of your accepted manuscript, and individual files for each figure and table included in your manuscript. You can send these in a zip folder if more convenient. Failure to provide these files may delay the processing of your proof.

on behalf of Professor Chris Chambers (Subject Editor)
openscience@royalsociety.org

Associate Editor Comments to Author

Three of the four original reviewers assessed the revised manuscript, and, happily, all are now satisfied. Reviewer 1 notes a couple of typos which can be addressed at the copyediting stage. Overall, I found this to be a very thorough and convincing revision, and I'm pleased to accept it without further changes. Congratulations!

Reviewer comments to Author:

Reviewer: 1

Comments to the Author(s)

I appreciate the authors' revision and careful consideration of my and the other referees' points, and as the authors stated in their response, this concrete initiative in the domain of cognitive modeling may do more to advance the debate regarding preregistration in that literature than would another more abstract or theoretical paper.

About the sentence on page 4 that ends with "areas of research", it is missing a period at the end of the sentence.

There is an extra closed parenthesis on page 21 where Steegen et al. are cited.

Reviewer: 2

Comments to the Author(s)

The changes made by the authors greatly improve the clarity and scholarship of the piece. I have no more feedback to give on this basis and believe this is probably in its final state and publishable.

Reviewer: 3

Comments to the Author(s)

All my comments are well addressed in the new version

Appendix A

This is a useful contribution to the literature that I think will improve work that uses cognitive and related models to estimate parameters that purport to measure mental constructs. The manuscript provides a novel preregistration template, which appears to be the first of its kind, applying a cognitive/mathematical model to behavioural data to estimate the model's parameters.

First, I should say that I have done very little cognitive modeling myself, so the expertise I bring to this review is more limited to various aspects of open science, preregistration, and experience with studies that focus on within-participant estimation with many trials per participant.

I think this is a needed contribution to the literature and I think the field will be appreciative of this work. My specific points about the manuscript, below, are largely about presentation and a few points that I think were missed.

“the added transparency of openly stating pre-existing knowledge will hopefully provide similar credibility, and at the very least enable the reader to put the results into context (Nosek et al. 2018).” I don't think that the preregistration skeptics will appreciate that phrase much, because shouldn't the Introduction to the resulting paper put the results into context anyway? Or maybe the authors mean something else, if so it could be clarified.

As the authors are aware (they indicate it with their citation of a Morey 2018 tweet), the overall value of preregistration for cognitive modeling is presently controversial, for example one paper was titled by its authors “Preregistration is redundant, at best”. For the purposes of this review, I will refer to such opinions as those of “preregistration skeptics”. I think the recent rise in preregistration skeptics makes it particularly important to ensure the writing about preregistration is clear and judicious. I think that it is NOT the job of this paper to settle those debates or even to take them head on. Clearly many researchers find preregistration valuable, making the provision of a template valuable.

Because this paper is about estimating parameters with a model, some of the preregistration skeptics' concerns are not applicable, and the authors point that out, which is useful. But here are a few comments:

The authors describe the purpose/benefits of preregistration variously, and these different benefits are scattered through the manuscript. It might be a good idea to bring them together. Appropriately, pretty early in the Introduction the researchers provide an extended motivation for preregistration including that the purpose is “to help researchers avoid fooling themselves into believing that their post-hoc explanations and explorations were a priori predictions and

predictions.” but at a later section they add that ”preregistration seems to be useful both for the a-priori justification of choices, and for the clarification of which choices do not have clear justifications.”

Some preregistration skeptics suggest that what is pre-planned is irrelevant to the inferences made after the data are in, because the theory is the theory regardless of what the researchers thought or didn't think about its predictions before the study was done. What this misses is that even putting aside publication bias and selective reporting, and even in those very few circumstances in which a theory is fully specified, enough for a second researcher to determine exactly how diagnostic the experiment was for adjusting their prior on that theory, fully evaluating that (how much to adjust one's prior and working out how strongly the theory predicted or didn't that result) is a huge amount of work, and the expertise required to do it is in very short supply. For many theories, there may be only a few dozen people in the world qualified to do it. Therefore one of the great benefits of preregistration is that it provides a credible signal of accurate prediction which has much lower cost to evaluate. That is, if a reader sees that another set of researchers made several preregistered predictions and those turned out to be correct, that's already a pretty good signal for the reader to increase the credibility of the theory that the researchers said they were using to make the prediction. Now, to be more confident in that, the reader should make some evaluation of whether it looks like the theory really did make those predictions, and that other theories didn't, which is the work that needs to be done without preregistration to *even get started* knowing whether to increase or decrease one's credence in the theory. Preregistration provides a shortcut, giving readers a rough sense already, even if it will be somewhat unreliable, it is a lot better than nothing. In a world with finite expert peer resources, we do benefit from this. I imagine some paper on preregistration in the literature already points out this benefit, although in my limited experience the preregistration skeptics seem to ignore it.

I noticed that the word “multiverse” doesn't appear in the manuscript, but the authors probably know that this is roughly a term for the phenomenon that the authors refer to as robustness checks, so they may want to mention this term there as well.

I really like the inclusion in the template of issues and contingency plans!

Minor writing-related comments

Up to page 6, aspects of the introduction are somewhat redundant, so I think the manuscript would have more impact if the authors tried to address this.

p.21 “If you are not interested in the parameters and are going straight to

statistical inference without estimating the parameters” I wasn’t sure what was meant by this, so I think many readers may have the same problem; probably it means if you just want to do something like a statistical test to assess whether the parameter values are higher in one condition or another - that might be worth clarifying.

About outlier exclusion, the manuscript suggests that “In purely experimental studies, this is usually focused at the level of entire participants. “ I think this is overstating things - in my experience, experimental studies frequently have outlier exclusion, for example trials with response times greater than a certain number, or greater than a certain number of standard deviations than the mean.

page 15, It’s weird to start a new section with “Moreover” and it’s not clear what the word means here.

Page 26: the word is “kinematogram”, not “kinetogram”. Incidentally, neither word appears in the paper cited.

“One potential critique of our preregistration template could be that the prompts are too open-ended. However, it has previously been found that a format with specific, open-ended questions is better at restricting researcher degrees of freedom than a purely open ended template (Veldkamp et al., 2018). “ This doesn’t seem to make sense. From the structure of the sentence, I was expecting the last thing, “open ended template” to be a contrast with “specific, open-ended questions”, but they both say they are open ended and I don’t understand what the difference is.

For the plate diagrams, it would be helpful to indicate what the subscripts, e.g., i and j , are iterating over.

For one of the plate diagrams, the caption includes “comparing the posterior distributions of the decision threshold parameters against the posterior predictive distributions for the optimal threshold”, in other words that two things are compared, but it’s not clear what is which. E.g., what shows the posterior predictive distributions for the optimal threshold? Is it the green region? In any case, please indicate what the green stuff is in the caption.

I hope the authors and editor find these comments to be useful.

Signed (I sign all my reviews),

Alex O. Holcombe

Appendix B

Review for: *Preregistration in Complex Contexts: A Preregistration Template for the Application of Cognitive Models*

Olivia Guest¹

¹*Donders Centre for Cognitive Neuroimaging, Radboud University, Nijmegen, The Netherlands*

March 16, 2021

Contents

Contents	1
1 General Comments	1
2 Abstract	2
3 Main Body	2
4 Final Comments	5
4.1 Acknowledgements	5
5 References	5

1 General Comments

This paper sets out to demonstrate through case study that preregistration can be an aid in mathematical psychology. With my review, I hope to provide a modeller’s perspective, since this community is being addressed, however I am not a mathematical psychologist under narrow definitions.

One main issue I stumble on is I am not entirely sure who the audience for this paper is, and some further clarification would be useful. This manuscript is directed at modellers — I assume — but I’m not sure if it engages with the way this community of researchers speaks about its own work. Relatedly, because modellers within cognitive science broadly are an established community of scientists, deploying language that the community uses formally or informally, e.g. words such as “complex” (which is used formally, often) or “experimental research” (which can be defined to include experiments run on computational models, or to be about just empirical data collected from participants) might also serve to confuse readers.

Another thing that might engage with modellers is to link your ideas for preregistration with related concepts or processes (Devezer, Navarro, Vandekerckhove, & Buzbas, 2020). Consideration of how the authors’ proposal for “preregistration of modelling” differs from the idea of a formal specification put forward in (Guest & Martin, 2021), or the idea of formal model comparison (Wills & Pothos, 2012) or specific open source modelling projects (Wills, O’Connell, Edmunds, & Inkster, 2017), would improve the paper. To be clear, I am sure they might differ, but these differences need to be explored in order to really get at things as well as to engage with what modellers currently do when they develop and compare their models to data and other models. Modelling is not a checklist and modellers constrain and explore their models in specific ways — engaging with this seems useful.

Another question I think should be addressed; do the authors want the various subfields of cognitive science to all move to deploying preregistration? Or are they proposing this template as an optional

step that modellers might select to carry out? Is preregistration even useful for modelling or are such checklists merely retarding the progress of theory-building? Is releasing code or discussion about our work the same as releasing useful high-level points? The cases of climategate (open emails) and Neil Fergusson (open code) are clear indications transparency and openness is not a solution but a constant dialogue — and can easily backfire. So what does that really mean for credibility when openness in those cases destroyed the credibility of the involved parties in many senses? How is openness a clear good here and how are modellers currently not open? I feel such general questions should be clearly answerable by the authors in this manuscript.

Most of my comments here and below are largely about the lack of details provided — so much of my feedback is about asking the authors to unpack what they mean, specify their assumptions, etc.

2 Abstract

In the abstract, what do the authors mean by “influential”? A bit of unpacking or another word might be more descriptive. Also, the same for “standard statistical analyses” — it might be useful to specify that you mean inferential stats because “assessing whether an experimental manipulation has an effect on a variable of interest” is so broad as to include computational cognitive modelling and general theorizing without the “inferential” or “data model” clarification.

Complex is a formalized word, so when used in expressions such as: “areas of research with more diverse hypotheses and more complex methods of analysis, such as cognitive modelling research within mathematical psychology” it fails to make sense. It is not true that a computational model is by definition more complex than a statistical model, if anything likely the opposite might be the case. To help with communication with the reader, the authors could perhaps define their formalism for complexity.

I think it might be good to give a little more on what the context is to say preregistration is under debate. Such details would help situate how this paper contributes to “the debate surrounding preregistration in cognitive modelling”, perhaps.

Given that the paper proposes to be about the titular “cognitive models”, as opposed to more specifically “models from mathematical psychology” the case study given is potentially misleading. For some readers, the example might involve high overheads, such as for people unfamiliar with the very specific techniques. The whole way of doing cognitive modelling might be unfamiliar to people who do other types of cognitive modelling. One solution might be to add more case studies or maybe change the title and framing to make it clear it’s directed at a specific community or way of modelling. Something seems very opaque, either in terms of a potentially confusing title or in terms of a confusion stemming from the framing of the paper or both. The authors could clarify and amend such issues to help the reader.

Another thing that might benefit from definition or some other form of clarity, is what is preregistration’s benefit itself in this context? If the benefit is empirically determined, i.e., we cannot do what preregistration provides without checking, then one case study is likely not convincing nor representative. If the role is as a principle, i.e., the authors believe preregistration is inherently good, then these assumptions should be stated and the context for such beliefs explained.

3 Main Body

I like the opening sentence as it frames the crisis as an issue as opposed to replication in and of itself. The rest of the first paragraph is confusing. It might benefit from establishing more clearly the relationship, if any, between preregistration, open science, and replication. Different scientists, especially cognitive modellers, might not hold the same assumptions as the authors.

On page 3, “Preregistration involves the specification of a researcher’s plans for a study, including hypotheses and analyses, typically before the study is conducted.” This is a good working definition. However, this is a strong claim that it “alleviate the effects of questionable research practices (QRPs) such as hypothesising after results are known (HARKing; Kerr, 1998) or p-hacking.” A researcher could construct a plan that does not exclude such events taking place, either accidentally or intentionally,

by making a plan that is too general or too theoretically unhinged. Tightening this is probably superficially easy.

Is “invalid” being used, on page 3, in the formal logical sense here? If so, this could be clarified.

Also, on page 3, where the authors say “most published studies in psychology claim to be confirmatory” might it be better to say “are perceived as” confirmatory and provide a definition of what that is? It appears to me that modellers, presumably the target audience, are not typically interested in, invested in, or even aware of, the distinction within experimental work that is proposed between confirming and exploring. Principally because cognitive modelling does not operate at a level where this distinction necessarily makes sense or applies.

Is the claim, on page 4, that “researchers may initially struggle to create preregistration documents that are appropriately detailed and justified” simply that more eyes are better? What do the authors mean by “simple hypotheses”? This again falls into the complex/simple distinction which tends to be a formal one — complexity is typically a formal concept, e.g., see Kolmogorov complexity for one definition.

The sentence, on page 5, “cognitive modelling, where researchers use mathematical models that are formal representations of cognitive processes to better understand human cognition” is another example of a phrase that is used in specific ways by modellers. A “mathematical model” might not be the same as a “computational model”, do the authors mean both here?

On page 5, some issues also arise because of how broadly preregistration is defined so far. It does stand to reason that the definition given for preregistration can easily encompass how cognitive modelling researchers already do their work. Recall: “Preregistration involves the specification of a researcher’s plans for a study, including hypotheses and analyses, typically before the study is conducted.” This can easily be seen to encompass a modeller who takes a pre-existing published model (and data) and runs the model *de novo* to test some hypotheses it generates (Guest & Martin, 2021). I assume the authors might want to rule out this interpretation, especially given their audience? Or are they being broad purposefully?

Page 6, “similar to how the development of general purpose preregistration templates and checklists have helped researchers to create well constrained preregistration documents for purely experimental studies”. Again, this is seemingly a very strong claim that probably needs empirical evidence at minimum to be made without any confusion.

On page 7, this sentence is a little confusing perhaps “cognitive models contain parameters that have psychological interpretations” — do the DVs and IVs in inferential statistics not have psychological interpretations without cognitive modelling? Something seems to be missing here to really hammer the point home of what cognitive modelling is. Maybe fixating on who the audience is might help.

On page 8, the phrase “the infrequent reuse of existing models”. Why is models falling out of favour a bad thing? If most/all models are wrong, but some are useful, then models not being used might be an indication they are more useless than average. Also, it’s a very strong claim to make even given the citation the authors use (compare with: Cooper & Guest, 2014; Guest, Caso, & Cooper, 2020; Guest & Martin, 2021). I think to really make this point one needs to define what use is and do a literature review, maybe even interviews.

Since researcher degrees of freedom is such a pivotal role played by preregistration in the authors’ opinion, they could define and unpack what it is earlier in the manuscript (or at least direct the reader that this will happen and when). Also, it might be useful to explain how this differs from (the largely neutral principle) of “modeller’s choice”, a basic principle in modelling. Relatedly, further down, the authors define “Model application, which will be the focus of our article, involves using an existing cognitive model to answer research questions about specific components of the underlying cognitive process.” This is very useful but also could be unpacked more.

On page 9, “Model application involves using cognitive models in a similar manner to statistical models (e.g., ANOVA), but with the assessments performed on the theoretically meaningful parameters estimated within the cognitive model rather than on the variables directly observed within the data.” Is this purposefully meant to exclude, e.g., a model that outputs data that is identical to participant data in kind, e.g., reaction time data? Is taking an existing model and reimplementing it (for examples,

see: Guest et al., 2020) “model application” in your modelling ontology? This is probably easy to fix by tightening the prose.

On page 10, “The use of preregistration is not limited to constraining the effects of common cognitive biases” — this is an empirical question. There is a plethora of research within the areas of judgement and decision-making, behavioural economics, etc., that might already exist to help back-up this claim. However, asserting this without evidence at all, is a little too strong.

On page 13, “Considering that one of the main advantages of cognitive modelling is generalisability based on substantive explanations for psychological phenomena (Busemeyer Diederich, 2010), one could argue that it is also a modeller’s degree of freedom to not prespecify motivations or justifications for different decisions.” This is why modeller’s choice needs to be discussed. Technically, and for all intents and purposes, there are infinite choices in many modelling paradigms. In other words, listing exhaustively all the alternatives at some levels of model development might be impossible since, for example, there are infinite infinitesimally small changes that can be made to a model to create a new other model, e.g., a deep neural network model might have more than 100 million parameters.

On page 15, “existing knowledge will hopefully provide similar credibility,” are the reader’s meant to infer that the authors think a main reason for preregistration is more credibility? If so, it might be useful to explain what credibility is lacking in modelling? Or if none is, and this is just additional credibility. Also, I assume a lack of credibility is identical to a lack of belief in the results and repercussions of an article by scientists. If not, this might need to be unpacked further, so readers can follow.

On page 15, “we believe that the goal of preregistration is not to prevent dishonesty or outright fraud, but to help researchers avoid fooling themselves into believing that their post-hoc explanations and explorations were a-priori predictions and decisions.” This is both a strong claim and presumably within a framework that somehow a priori hypotheses are different to a posteriori ones in some deep way (e.g., see Kataria, 2016). This could benefit from being unpacked and explained to the modellers being addressed, or indeed any readers. Especially so since many in mathematical psychology are Bayesians in multiple senses.

On page 17, if you already have the code how are you preregistering? How can you check your code works if not by already running it against previous data, to have module testing, etc.? The authors here need to deeply engage with the point in my paper they cite, (Cooper & Guest, 2014). This paper explains how code is not sufficient to constrain things in a meaningful way without a specification.

On page 18, “Rather, we believe that our preregistration template may be a useful tool to help ensure robustness and transparency in model application studies”. Is this also not (at least ultimately) an empirical claim? I’m not entirely sure what robustness is in this context, and perhaps it might benefit from description or definition?

On page 19, “Any post-hoc addition or modulation is then clearly exploratory rather than confirmatory research.” What if the modellers are using an existing model but while writing the preregistration did not realize they misunderstood something in the original model? This is extremely common in modelling, especially when re-implementing models (Cooper & Guest, 2014; Guest et al., 2020).

On page 25, in fact this whole section on a computational model and a specific experimental set-up, I am not familiar with. So my comments here must be taken with a pinch of salt. As with all modelling, in my experience, it’s very hard to have an informed opinion without some playing around with the implementation, if not building it from scratch.

On page 25, “The assessment in Evans & Brown (2017) was rather qualitative and not very rigorously defined, making this a good opportunity to show how preregistration in cognitive modelling can add rigor and transparency in situations with many potential researcher degrees of freedom.” As we know from modelling and theorizing, such comparisons can easily be spun, and often are, without controls in place, to favour a specific outcome. Hence, why might it be good to consider a contrast condition without preregistration perhaps?

On pages 27-28, why do the authors choose to define the hypotheses qualitatively? I don’t know what is more appropriate, but a formal definition with reference to the expected effects might be possible too here and useful.

On page 31, “Nevertheless, deviations from a preregistration should always be possible, for example

if the researcher gained important knowledge in between writing the preregistration and analysing the data”. Doesn’t this always happen with modelling, and so the modeller will always be forced to deviate? This might not be a problem, but needs to be discussed.

On page 36, “No preregistration is perfect, and using an overly general template – or one designed for a different field of research – is likely better than using no template or eschewing preregistration entirely.” What does better mean or imply here? Either way this needs unpacking because it’s a very strong claim and does not seem superficially true (if at all).

4 Final Comments

Overall I think this paper and concept need a lot more work to actually be appealing in a meaningful way to the communities that seem to be addressed. Even though I remain unconvinced of the scientific value of preregistration here, clarifying all the points I mentioned above are likely to heighten the contradictions (for people like me who disagree) but also likely to further engage whoever the intended audience is (it is not fully clear to me which exact communities are being addressed).

Ultimately, it will fall to modellers and their communities if they wish to adopt a checklist. In my opinion, such checklists, if not couched and framed properly, can easily debase and devalue modelling work making it inherently unscientific by unhinging models from the context via a mindless box ticking exercise. This is likely possible to be tempered, controlled for, and even avoided, I think, if the authors try to be clearer with their ideas and prose.

I hope my comments cause pause for thought and further exploration of their own ideas and proposals which seem to be prescriptively stated, again something that can be side-stepped with more careful language and perhaps deeper conceptual analysis and/or engineering.

4.1 Acknowledgements

I would like to thank Angus Inkster for the invaluable feedback and useful discussions regarding this review.

5 References

- Cooper, R. P., & Guest, O. (2014). Implementations are not specifications: Specification, replication and experimentation in computational cognitive modeling. *Cognitive Systems Research*, *27*, 42–49.
- Devezer, B., Navarro, D. J., Vandekerckhove, J., & Buzbas, E. O. (2020). The case for formal methodology in scientific reform. *bioRxiv*.
- Guest, O., Caso, A., & Cooper, R. P. (2020). On simulating neural damage in connectionist networks. *Computational brain & behavior*, *3*(3), 289–321.
- Guest, O., & Martin, A. E. (2021). How computational modeling can force theory building in psychological science. *Perspectives on Psychological Science*. Retrieved from <https://doi.org/10.1177/1745691620970585> (PMID: 33482070) doi: 10.1177/1745691620970585
- Kataria, M. (2016). Confirmation: What’s in the evidence? *Journal of Behavioral and Experimental Economics*, *65*, 9–15.
- Wills, A. J., O’Connell, G., Edmunds, C. E., & Inkster, A. B. (2017). Progress in modeling through distributed collaboration: Concepts, tools and category-learning examples. In *Psychology of learning and motivation* (Vol. 66, pp. 79–115). Elsevier.
- Wills, A. J., & Pothos, E. M. (2012). On the adequacy of current empirical evaluations of formal models of categorization. *Psychological bulletin*, *138*(1), 102.

Appendix C

Prof Chris Chambers
Associate Editor
Royal Society Open Science

RE: Revision

Dear Prof Chambers,

Thank you for the opportunity to revise our manuscript titled “Preregistration in Diverse Contexts: A Preregistration Template for the Application of Cognitive Models”. We would like to thank yourself and the reviewers for the helpful and constructive feedback, and we have attempted to address all of the comments within our revision. In the “responses” section below, we have responded to each comment in detail. In the manuscript, the changes made are in red, we did not include sentences and words that were deleted or replaced.

In your evaluation, you highlighted specifically the concerns that 1) the audience could be made clearer, and 2) that the template may be seen to be redundant. Concerning these specific issues, we would like to summarise the relevant responses here.

Audience

The intended audience is very much interdisciplinary, and this is interdisciplinary work -- as is reflected in the different key areas of expertise of the authors. We think that this paper is relevant to both an open science and a cognitive modelling audience (though note that our intention is not to convince cognitive modelling researchers who may be critical of preregistration that they should be using preregistration, but rather to provide a template for cognitive modelling researchers interested in preregistering their model application studies). We realise that this is a difficult audience to aim for, and we think that this has prompted some of the comments raised by the reviewers. In responding to many of the points made by Reviewer 1 and 2 in particular, we hope we have clarified this. Furthermore, we have

tried to make it clearer in the introduction that the project and its intended audience are interdisciplinary:

“The goal of this paper is twofold: 1) to introduce a template that modellers can critique, improve upon, and use if they want to; 2) to show Open Science advocates another extension of their existing templates and how the principles they try to further may also be applied in model application.¹ We realise that there is a disconnect between the two communities: Open Science advocates believe that their Open Science principles are broadly applicable, whereas modellers by and large believe that these principles are not applicable to cognitive modelling. Overall, this debate has been held in the abstract, which can be useful, but should be complemented with concrete proposals. Our template for preregistration in model application is one such concrete proposal.”

Importance

Regarding the concern about the redundancy or importance of the preregistration template, as expressed by Reviewer 4, Comment 1: we believe that it is somewhat misleading to say that our template is redundant simply because it overlaps with the standard OSF checklist. Within our manuscript, we are very clear that our template is an *extension* of the existing OSF checklist -- as many of the other more specific preregistration templates and checklists have been (e.g., Flannery, 2018; Haven & Grootel, 2019; Paul et al., 2021; Kirtley et al., 2021). Furthermore, our manuscript only focuses on our extensions of the standard template for use in model application, i.e., in our manuscript we are not repeating content from the standard OSF checklist.

While we see Reviewer 4's point that our template is not completely unique from all other templates, we believe that questioning the value of our template for this reason would be similar to questioning the value of a new model because it extends from an existing model (e.g., questioning the value of Ratcliff's 1978 diffusion model -- one of the most influential models in mathematical psychology -- because it only adds a single parameter to Stone's 1960 diffusion model). Instead, we believe that template

¹ This latter goal is also important given that many Open Science advocates have been involved in the debates surrounding this topic.

extensions should be assessed on *how much value the extensions add* (e.g., similar to how model extensions are assessed) for those who wish to preregister model application studies, but may not be sure how to do so.

We believe that there are several relevant arguments for the usefulness of our template, and more generally, the usefulness of specific templates that extend from the existing standard OSF checklist, including:

1) Specific templates make the application of preregistration more straightforward for researchers. They don't have to reinvent the wheel every time, and they can build on something that other researchers in their area have used and hopefully improved on before.

2) Specific templates are helpful for standardisation. If every modeller wanting to preregister a model application study tried to use the standard OSF checklist for the purpose of model application, we would have many different versions trying to do the same thing, but likely doing it in very different ways. Our proposed template would at least give one common starting point for model application in cognitive modelling.

3) Specific templates may encourage researchers to be exhaustive in their preregistrations. Using a "standard" preregistration template, you run the risk of leaving things out.

4) As argued by Reviewer 2 Comment 4: a preregistration can serve as a basic but credible signal of accurate prediction for readers not familiar with a field or theory, which may be particularly useful for niche and intricate theories and areas of research, such as much of cognitive modelling. Therefore, lowering the threshold for preregistration by proposing a specific template for model application in cognitive modelling seems worthwhile and important.

5) Most importantly: Given the contentious debate surrounding preregistration in this particular area, a template for a part of cognitive modelling also represents a counter-argument to the idea that preregistration in cognitive modelling is simply infeasible in all areas and categories of cognitive modelling.

6) Finally, the preprint of this manuscript has already been cited 7 times (including Corker, 2021) and has been downloaded more than 500 times. This indicates a general interest in this type of template in the broader research community. We are also aware of at least one preregistration that uses our template: <https://osf.io/7gt45/>.

Overall, we believe the question of redundancy/importance is a question of A) are specific templates potentially useful?, and B) does our specific template provide additional material that may assist researchers in preregistering model application studies (at least to a similar extent as other previous extensions)? We believe that A) is the case, and that our template achieves B) to at least a similar extent as previous extensions, and we hope that our extensive responses and revisions also convince the reviewers and editor of this. We hope that you will find our manuscript suitable for publication in *Royal Society Open Science*.

Sincerely,

Sophia Crüwell

University of Cambridge

slbc2@cam.ac.uk

Nathan J. Evans

University of Queensland

nathan.j.evans@uon.edu.au

Responses

Reviewer 1 (Olivia Guest)	This paper sets out to demonstrate through case study that preregistration can be an aid in mathematical psychology. With my review, I hope to provide a modeller’s perspective, since this community is being addressed, however I am not a mathematical psychologist under narrow definitions.	Thank you for your detailed and helpful review. We appreciate your expertise as a modeller and the insight that you have provided, and we hope we have appropriately addressed your questions and comments in our revised manuscript. However, we would like to note here that our aim was *not* to “demonstrate through case study that preregistration can be an aid in mathematical psychology”, and rather to provide a concrete proposal of a preregistration template for model application for those who are interested in preregistration (as pointed out by Reviewer 2 in Reviewer 2, Comment 2, and discussed in our response to Reviewer 1, General Comment 3), and show that preregistration is possible for at least some types of cognitive modelling. We have attempted to make our aim clearer in the revised version of the manuscript.
Reviewer 1, General Comment 1	One main issue I stumble on is I am not entirely sure who the audience for this paper is, and some further clarification would be useful. This manuscript is directed at modellers — I assume — but I’m not sure if it engages with the way this community of researchers speaks about its own work. Relatedly, because modellers within cognitive science broadly are an established community of scientists, deploying language that the community uses formally or informally, e.g. words such as “complex” (which is used formally, often) or “experimental research” (which can be defined to include experiments run on computational models, or to be about just empirical data	Thank you for bringing this up! As mentioned in our response to the editor, our intention was to direct our manuscript at *both* the modelling and open science communities. Specifically, we believe that there are researchers (both “modellers”, and those who are interested in applying models to their data, but may not consider themselves to be “modellers” per se) who are interested in preregistering their model application studies, as well as open science researchers who are interested in making preregistration more accessible to researchers in different areas. We hope that the readership of our article will be both groups of researchers, and therefore, we attempted to make the language, background literature review, etc. all as accessible as possible to people from both areas. However, attempting to make an article “interdisciplinary” can be a difficult goal, and we’re very thankful

	collected from participants) might also serve to confuse readers.	for all of the instances that you brought up where our language may have been confusing to those within the modelling community. We have attempted to clarify these points throughout, and we hope that our revisions have made our manuscript clearer. For example, we have replaced the word “complex” with “diverse” or “intricate” in most contexts, as we believe this better conveys our intended meaning. We have also added the following footnote to the first use of the term “experimental research”: “Note that while mathematical modelling research often involves experimental tests, when we talk of (purely) experimental research here and below, we refer to experimental research that uses standard inferential statistical methodology.” (p.6)
Reviewer 1, General Comment 2	Another thing that might engage with modellers is to link your ideas for preregistration with related concepts or processes (Devezer, Navarro, Vandekerckhove, & Buzbas, 2020). Consideration of how the authors’ proposal for “preregistration of modelling” differs from the idea of a formal specification put forward in (Guest & Martin, 2021), or the idea of formal model comparison (Wills & Pothos, 2012) or specific open source modelling projects (Wills, O’Connell, Edmunds, & Inkster, 2017), would improve the paper. To be clear, I am sure they might differ, but these differences need to be explored in order to really get at things as well as to engage with what modellers currently do when they develop and compare their models to data and other models. Modelling is not a checklist and modellers constrain and explore their	Thank you for bringing up these points. We added the following pieces of text to try and better unpack these issues: “Model development involves the initial development of a model, or the extension/reduction of an existing model to create a new model, which is often an iterative, exploratory process and thus not well suited to preregistration (for a discussion on how open theorising can help constrain inference, see Guest & Martin 2020).” (p.10) “Model comparison involves directly contrasting multiple models on their ability to account for a set of empirical data, which is usually performed quantitatively through model selection methods (e.g., AIC, Akaike 1974; BIC, Schwarz 1978; see Evans 2019a for a discussion). These latter two categories may also benefit from preregistration (though see Wills & Pothos, 2012 for an alternative approach for model comparison: formal model comparison).” (p.10) “Well-documented code can also play a complementary role within preregistration (and for open modelling in general, see e.g. Wills

	models in specific ways — engaging with this seems useful.	et al. 2017 for a discussion of distributed collaboration).” (p.20)
Reviewer 1, General Comment 3	Another question I think should be addressed; do the authors want the various subfields of cognitive science to all move to deploying preregistration? Or are they proposing this template as an optional step that modellers might select to carry out? Is preregistration even useful for modelling or are such checklists merely retarding the progress of theory-building? Is releasing code or discussion about our work the same as releasing useful high-level points? The cases of climategate (open emails) and Neil Fergusson (open code) are clear indications transparency and openness is not a solution but a constant dialogue — and can easily backfire. So what does that really mean for credibility when openness in those cases destroyed the credibility of the involved parties in many senses? How is openness a clear good here and how are modellers currently not open? I feel such general questions should be clearly answerable by the authors in this manuscript. Most of my comments here and below are largely about the lack of details provided — so much of my feedback is about asking the authors to unpack what they mean, specify their assumptions, etc.	Thank you for raising this important point. We would like to be very clear that we certainly do not wish to force modellers who disagree with preregistration to use preregistration, and in terms of the modelling community, our template is directed at those who are either on the fence about preregistration for model application studies, or who wish to preregister their model application studies but are unsure how to do so. To us, preregistration is an optional step that is the choice of the researcher. Regarding the goal of our manuscript, as noted by Reviewer 2 (Comment 2), our intention for this manuscript is to provide a concrete proposal of a preregistration template for model application for those who are interested in preregistration, rather than to debate the merits for preregistration. Importantly, we do not wish to turn our article into an abstract philosophical debate on the pros and cons of preregistration in cognitive modelling, as we believe that several of these pieces already exist arguing each side, and that a more useful next step is to provide concrete proposals that advocates can extend and critics can critique. Furthermore, this preregistration template is not aimed at the “theory building” end of the modelling spectrum (i.e., what we label as “model development”), or the assessment and/or comparison of different competing models that propose fundamentally different underlying processes (i.e., what we label model evaluation and model comparison, respectively). However, we agree that the goal of our article may have been somewhat unclear, as well as what type of “cognitive modelling” we were providing a preregistration template for. Therefore, we have attempted to clarify this in the

		revised manuscript in several places, though particularly on page 3 where we now state: “The goal of this paper is twofold: 1) to introduce a template that modellers can critique, improve upon, and use if they want to; 2) to show Open Science advocates another extension of their existing templates and how the principles they try to further may also be applied in model application. We realise that there is a disconnect between the two communities: Open Science advocates believe that their Open Science principles are broadly applicable, whereas modellers by and large believe that these principles are not applicable to cognitive modelling. Overall, this debate has been held in the abstract, which can be useful, but should be complemented with concrete proposals. Our template for preregistration in model application is one such concrete proposal.” We have also tried to make the scope of the template and of the paper introducing it clearer, see your comment 1 and: “...how it might be implemented for model application (more on this in the subsection on cognitive modelling; Crüwell, Stefan, & Evans, 2019) in...” (p.3) In the same section, we also added the following footnote: “While we think that this work may be applicable to all model application studies (more on this below) across cognitive modelling, the main focus of this paper are cognitive models within mathematical psychology. Our proposed template may not be well or immediately suited for cognitive models in other areas, but we believe that it may also serve as a useful starting point there.” (p.3)
--	--	---

Reviewer 1, Abstract, Comment 1	In the abstract, what do the authors mean by “influential”? A bit of unpacking or another word might be more descriptive. Also, the same for “standard statistical analyses” — it might be useful to specify that you mean inferential stats because “assessing whether an experimental manipulation has an effect on a variable of interest” is so broad as to include computational cognitive modelling and general theorizing without the “inferential” or “data model” clarification.	Thank you for noting this, these points were previously unclear. We have now replaced “influential” with “increasingly used”, and added “inferential” to “statistical analyses”.
Reviewer 1, Abstract, Comment 2	Complex is a formalized word, so when used in expressions such as: “areas of research with more diverse hypotheses and more complex methods of analysis, such as cognitive modelling research within mathematical psychology” it fails to make sense. It is not true that a computational model is by definition more complex than a statistical model, if anything likely the opposite might be the case. To help with communication with the reader, the authors could perhaps define their formalism for complexity.	Good point; as discussed in Reviewer 1, General Comment 1, we have replaced “complex” with “intricate” here, and have avoided using the word “complex” throughout the revised manuscript, as we were not intending to refer to this formal definition of the word.
Reviewer 1, Abstract, Comment 3	I think it might be good to give a little more on what the context is to say preregistration is under debate. Such details would help situate how this paper contributes to “the debate surrounding preregistration in cognitive modelling”, perhaps.	Good point; we agree that this issue should be unpacked further, though we believe that properly unpacking this issue in the abstract would make it too lengthy. Therefore, instead of adding this to the abstract, we unpack this issue further in the Introduction on page 5/6, stating: “Some question the general usefulness of preregistration in areas of psychological research with more

		diverse hypotheses and more intricate analyses, e.g. pointing out the exploratory nature of model development or focussing on the development of strong theory (MacEachern & Van Zandt 2019; Szollosi et al. 2019; see also the “Cognitive Modelling” and “Issues and Peculiarities in Preregistering Model Application” sections where we address several common challenges and misconceptions). Others believe that preregistration could still serve an important purpose in constraining researcher degrees of freedom in other categories of cognitive modelling, and even where there is strong theory (Wagenmakers & Evans, 2018; Lee et al., 2019; Crüwell et al., 2019).” (p.6) However, as discussed in our response to Reviewer 1, General Comment 3, our main goal isn’t to directly debate the usefulness of preregistration in cognitive modelling, but instead to provide a concrete template for those who wish to perform preregistration.
Reviewer 1, Abstract, Comment 4	Given that the paper proposes to be about the titular “cognitive models”, as opposed to more specifically “models from mathematical psychology” the case study given is potentially misleading. For some readers, the example might involve high overheads, such as for people unfamiliar with the very specific techniques. The whole way of doing cognitive modelling might be unfamiliar to people who do other types of cognitive modelling. One solution might be to add more case studies or maybe change the title and framing to make it clear it’s directed at a specific community or way of modelling. Something seems very opaque, either in terms of a potentially confusing	That’s a good point, thank you. The focus of this template and paper is on model application in “cognitive modelling research within mathematical psychology” (in the abstract). However, we believe that there are at least some overlapping principles across cognitive modelling in different areas. Accordingly, we have left the title as “the application of cognitive models”, as we believe that our template could be more broadly of interest to those who perform model application with cognitive models in areas other than mathematical psychology (particularly those who may work in completely different areas, such as clinical psychology, but may implement models traditionally from mathematical psychology as part of their work). We have tried to make it clear throughout our manuscript that our template does not extend to other types of cognitive modelling, such as model evaluation and model development. While we believe that

	title or in terms of a confusion stemming from the framing of the paper or both. The authors could clarify and amend such issues to help the reader.	clarifying these points further is probably too specific for the abstract, we have added a footnote to the first part of the introduction, where we discuss the point that our template is for “cognitive modelling studies within mathematical psychology” (p. 3): “While we think that this work may be applicable to model application studies (more on this below) across cognitive modelling, the main focus of this paper are cognitive models within mathematical psychology. Our proposed template may not be well or immediately suited for cognitive models in other areas, but we believe that it may also serve as a useful starting point there.” (p.3)
Reviewer 1, Abstract, Comment 5	Another thing that might benefit from definition or some other form of clarity, is what is preregistration’s benefit itself in this context? If the benefit is empirically determined, i.e., we cannot do what preregistration provides without checking, then one case study is likely not convincing nor representative. If the role is as a principle, i.e., the authors believe preregistration is inherently good, then these assumptions should be stated and the context for such beliefs explained.	This is a good point, and as discussed in Reviewer 1, General Comment 3, we feel that this may be due to a previous lack of clarity in our goal. Importantly, we are not trying to make a strong argument about preregistration here, or attempting to convince critics that they should be using preregistration. We have attempted to further clarify this within the revised manuscript. And we have now specified in the main body of the paper that the major aim of this paper is to add a concrete proposal to an abstract debate (see our response to the Editor’s comment). We have further slightly amended the relevant sentence of the abstract to read: “More broadly, we hope that our discussions and concrete proposals constructively advance the mostly abstract current debate surrounding preregistration in cognitive modelling, and provide a guide for how preregistration templates may be developed in other diverse or intricate research contexts.” (p.2)
Reviewer 1, Main Body, Comment 1	I like the opening sentence as it frames the crisis as an issue as opposed to replication in and of itself. The rest of the first paragraph is confusing. It might	Good point; we agree that the first paragraph may have previously come across as an authoritative detailing of the benefits of open science practices, which was not our intention. Instead, this first paragraph is intended to be

	benefit from establishing more clearly the relationship, if any, between preregistration, open science, and replication. Different scientists, especially cognitive modellers, might not hold the same assumptions as the authors.	descriptive rather than normative, and to merely discuss what has occurred in previous research in the area of open science. Therefore, we have attempted to make this clearer throughout the first paragraph, clarifying that the “reform” practices that are often called “open science practices” are a subset of what may be called “open science”, and that these reforms were proposed in an attempt to increase rigour and replicability (whether this attempt was or can be successful is a different topic for debate, and an empirical question, as we mention on p.21).
Reviewer 1, Main Body, Comment 2	On page 3, “Preregistration involves the specification of a researcher’s plans for a study, including hypotheses and analyses, typically before the study is conducted.” This is a good working definition. However, this is a strong claim that it “alleviate the effects of questionable research practices (QRPs) such as hypothesising after results are known (HARKing; Kerr, 1998) or p-hacking.” A researcher could construct a plan that does not exclude such events taking place, either accidentally or intentionally, by making a plan that is too general or too theoretically unhinged. Tightening this is probably superficially easy.	Thanks for pointing this out. We have amended this sentence and claim: “It has been suggested that preregistration can help to make the research process more transparent, to constrain researcher degrees of freedom (i.e., undisclosed flexibility in study design, data collection, and/or data analysis; Simmons, Nelson, & Simonsohn, 2011), and to help alleviate the effects of questionable research practices (QRPs) such as hypothesising after results are known (HARKing; Kerr, 1998) or p-hacking (Nosek et al., 2018; Munafò et al., 2017; Wagenmakers et al., 2012).” (p.4)
Reviewer 1, Main Body, Comment 3	Is “invalid” being used, on page 3, in the formal logical sense here? If so, this could be clarified.	We have changed “invalid” to “unwarranted” and “interpretation of results” to “inference”.
Reviewer 1, Main Body, Comment 4	Also, on page 3, where the authors say “most published studies in psychology claim to be confirmatory” might it be better to	We agree that the confirmatory/exploratory dichotomy is not necessarily a useful one in all situations, but chose to use these terms

	say “are perceived as” confirmatory and provide a definition of what that is? It appears to me that modellers, presumably the target audience, are not typically interested in, invested in, or even aware of, the distinction within experimental work that is proposed between confirming and exploring. Principally because cognitive modelling does not operate at a level where this distinction necessarily makes sense or applies.	to be in line with previous literature. However, we have also added a footnote to clarify our use of this unfortunate but standard dichotomy: “In discussions surrounding preregistration, research is often dichotomised into strictly exploratory and strictly confirmatory research. “Confirmatory” research here means research in which hypotheses and analyses were planned before the start of the study. This is arguably not a useful dichotomy (Scheel et al., 2020; Szollosi & Donkin, 2019), and exploratory and confirmatory research are likely part of a broader spectrum. Nevertheless, we will use these terms in the context of describing the debate surrounding preregistration and cognitive modelling to reflect the language and concepts used by both proponents and opponents of preregistration in cognitive modelling.” (p.4) Also, we would like to emphasise again that the template proposed here is only meant for what we term “model application”, and not for other types of cognitive modelling, such as “model development”.
Reviewer 1, Main Body, Comment 5	Is the claim, on page 4, that “researchers may initially struggle to create preregistration documents that are appropriately detailed and justified” simply that more eyes are better? What do the authors mean by “simple hypotheses”? This again falls into the complex/simple distinction which tends to be a formal one — complexity is typically a formal concept, e.g., see Kolmogorov complexity for one definition.	Our point on page 4 is that the registered reports process has an extra layer of “checking” (i.e., the peer review of the preregistration), and is less our claim than one made in the previous literature. We have replaced both instances of “simple” in the relevant sentence, which now reads: “Therefore, they are applicable to studies where researchers are interested in testing straightforward hypotheses, such as whether an experimental manipulation has an effect on a variable of interest, with standard analysis tools, such as a null hypothesis significance test on an interaction term within an ANOVA.” (p.5/6)

Reviewer 1, Main Body, Comment 6	The sentence, on page 5, “cognitive modelling, where researchers use mathematical models that are formal representations of cognitive processes to better understand human cognition” is another example of a phrase that is used in specific ways by modellers. A “mathematical model” might not be the same as a “computational model”, do the authors mean both here?	We have specified that we mean both mathematical and computational models here.
Reviewer 1, Main Body, Comment 7	On page 5, some issues also arise because of how broadly preregistration is defined so far. It does stand to reason that the definition given for preregistration can easily encompass how cognitive modelling researchers already do their work. Recall: “Preregistration involves the specification of a researcher’s plans for a study, including hypotheses and analyses, typically before the study is conducted.” This can easily be seen to encompass a modeller who takes a pre-existing published model (and data) and runs the model de novo to test some hypotheses it generates (Guest & Martin, 2021). I assume the authors might want to rule out this interpretation, especially given their audience? Or are they being broad purposefully?	Thank you for pointing this out. We have clarified our definition of preregistration (see also Reviewer 2, Comment 3 and Minor Comment 1; Reviewer 3 Additional Comment 1), and included the fact that most preregistrations are time-stamped documents. We agree that the example you give is something that could form the basis of a solid preregistration. However, without clearly specified, written, and ideally timestamped documentation of the analysis process, we would argue that this itself does not constitute a preregistration.
Reviewer 1, Main Body, Comment 8	Page 6, “similar to how the development of general purpose preregistration templates and checklists have helped	Good point; we have made two changes to address this. First, we agree that this comment may have sounded somewhat stronger than we originally intended, and we have attempted to reword it

	researchers to create well constrained preregistration documents for purely experimental studies”. Again, this is seemingly a very strong claim that probably needs empirical evidence at minimum to be made without any confusion.	based on the limited amount of empirical evidence. Second, we have included a further reference to Bakker et al (2020), who evaluated the quality of less constrained vs more constrained preregistration templates.
Reviewer 1, Main Body, Comment 9	On page 7, this sentence is a little confusing perhaps “cognitive models contain parameters that have psychological interpretations” — do the DVs and IVs in inferential statistics not have psychological interpretations without cognitive modelling? Something seems to be missing here to really hammer the point home of what cognitive modelling is. Maybe fixating on who the audience is might help.	Good point. The previous version of this sentence lacked clarity and we have amended this to read: “cognitive models contain parameters that directly reflect psychological constructs”. (p.8) Specifically, we do not believe that DVs in inferential statistics directly reflect psychological constructs, whereas cognitive models do, as parameters relate to specific parts of the underlying cognitive process.
Reviewer 1, Main Body, Comment 10	On page 8, the phrase “the infrequent reuse of existing models”. Why is models falling out of favour a bad thing? If most/all models are wrong, but some are useful, then models not being used might be an indication they are more useless than average. Also, it’s a very strong claim to make even given the citation the authors use (compare with: Cooper & Guest, 2014; Guest, Caso, & Cooper, 2020; Guest & Martin, 2021). I think to really make this point one needs to define what use is and do a literature review, maybe even interviews.	Good point; we agree that models falling out of favour is not necessarily a bad thing, and therefore, we have removed this phrase.

Reviewer 1, Main Body, Comment 11	Since researcher degrees of freedom is such a pivotal role played by preregistration in the authors' opinion, they could define and unpack what it is earlier in the manuscript (or at least direct the reader that this will happen and when). Also, it might be useful to explain how this differs from (the largely neutral principle) of "modeller's choice", a basic principle in modelling. Relatedly, further down, the authors define "Model application, which will be the focus of our article, involves using an existing cognitive model to answer research questions about specific components of the underlying cognitive process." This is very useful but also could be unpacked more.	Good point; we agree that the previous version of the manuscript may have been lacking in some linking. In the introduction, when we explicitly introduce and define researcher degrees of freedom, we have now added a reference to the later section on "Researcher Degrees of Freedom in Model Application", so that readers will know where these ideas will be further unpacked. We have also added the following adjustment on p.8: "(though freedom in modelling is not always viewed as a negative, see e.g. MacEachern & Van Zandt, 2019)", in order to make it clear that freedom in modelling is certainly not always a bad thing. Regarding the last point, we apologise for the lack of clarity. The remainder of this paragraph was intended to unpack the idea of model application in more detail, and we now clearly state this in the revised manuscript.
Reviewer 1, Main Body, Comment 12	On page 9, "Model application involves using cognitive models in a similar manner to statistical models (e.g., ANOVA), but with the assessments performed on the theoretically meaningful parameters estimated within the cognitive model rather than on the variables directly observed within the data." Is this purposefully meant to exclude, e.g., a model that outputs data that is identical to participant data in kind, e.g., reaction time data? Is taking an existing model and reimplementing it (for examples, see: Guest et al., 2020) "model application" in your modelling ontology? This is probably easy to fix by tightening the prose.	Good point (see also Reviewer 4, Comment 5); we agree that these definitions may have been unclear in the previous version of the manuscript. Importantly, we do not believe that the model itself defines the "type" of cognitive modelling that is being done. Rather, how the model is being used determines the type of cognitive modelling, and a single model can certainly be used for different types of cognitive modelling in different contexts. Regarding the specific examples in the review, using a model to simulate data is not model application, but model application could be performed in a simulation study using data simulated from a model. In the specific example of Guest et al (2020), the existing model is reimplemented with the purpose (if we understood this correctly) of assessing how well the model performs, which would be an instance of "model evaluation" (a more detailed discussion of this can be found in Cruwell et al., 2019).

		We have attempted to clarify the point that the model does not determine the type of modelling being done -- rather, its usage does -- in the main body of the text: “Model application, which will be the focus of our article, involves using an existing cognitive model to answer research questions about specific components of the underlying cognitive process -- note again that it is the intended purpose of using the model that determines the category of cognitive modelling research.” (p.10)
Reviewer 1, Main Body, Comment 13	On page 10, “The use of preregistration is not limited to constraining the effects of common cognitive biases” — this is an empirical question. There is a plethora of research within the areas of judgement and decision-making, behavioural economics, etc., that might already exist to help back-up this claim. However, asserting this without evidence at all, is a little too strong.	Thank you for this good point. We neither want to nor can make strong claims about preregistration in this context. We have changed this sentence as follows: “The intended use of preregistration is not limited to potentially constraining the effects of common cognitive biases -- preregistration may also help to organise and streamline a research workflow.” (p.11)
Reviewer 1, Main Body, Comment 14	On page 13, “Considering that one of the main advantages of cognitive modelling is generalisability based on substantive explanations for psychological phenomena (Busemeyer Diederich, 2010), one could argue that it is also a modeller’s degree of freedom to not prespecify motivations or justifications for different decisions.” This is why modeller’s choice needs to be discussed. Technically, and for all intents and purposes, there are infinite choices in many modelling paradigms. In other	This is a really important point. We agree that this sentence/paragraph could be understood as being about any category of cognitive modelling, including model development. However, that is not our intention -- our overall point is very much only about model application. We do use an example from model evaluation to make a broader point, which may have been misleading. This is now clearly labelled, and we have clarified the categories addressed in various sentences, including: “First and foremost, it is important that the choices made at all stages of a model application study are justified and motivated, which is often not the case in cognitive modelling studies that use model application (Dutilh et al.,

	words, listing exhaustively all the alternatives at some levels of model development might be impossible since, for example, there are infinite infinitesimally small changes that can be made to a model to create a new other model, e.g., a deep neural network model might have more than 100 million parameters.	2018; Starns et al., 2019).” (p.14) As well as adding a separate point specifically for model application: “In model application, a lack of specification and motivation may allow a researcher to change the form of a model depending on whether the model applied to their data results in the difference between groups they were looking for.” (p.15)
Reviewer 1, Main Body, Comment 15	On page 15, “existing knowledge will hopefully provide similar credibility,” are the reader’s meant to infer that the authors think a main reason for preregistration is more credibility? If so, it might be useful to explain what credibility is lacking in modelling? Or if none is, and this is just additional credibility. Also, I assume a lack of credibility is identical to a lack of belief in the results and repercussions of an article by scientists. If not, this might need to be unpacked further, so readers can follow.	Good point. We have changed that sentence (see also Reviewer 3, Additional Comment 5) to be: “Although such a preregistration will not function in quite the same way as it is meant to with original data, the additional transparency of openly stating all pre-existing knowledge of the specific secondary data sets that one plans to analyse will hopefully provide similar added credibility, and at the very least provide the reader with more context on the research process that produced the results presented in the corresponding paper (Nosek et al., 2018).” (p.17) Part of the motivation behind preregistration is transparency in that everyone can see what decisions were made when. See also the point made by Reviewer 2 (Comment 4), which we have now included in the manuscript on p.4: “This is not only helpful for the researchers themselves, but may also serve as a rudimentary but credible signal of accurate prediction for readers not familiar with a field or theory, which may be particularly useful for niche and intricate theories and areas of research.”
Reviewer 1, Main Body, Comment 16	On page 15, “we believe that the goal of preregistration is not to prevent dishonesty or outright fraud, but to help researchers avoid fooling themselves into believing that their post-hoc	While we see the reviewer’s point, we do not believe that this is a strong claim, as we only say that we believe that this is the *goal* of preregistration. Specifically, we do not claim that (1) preregistration necessarily achieves this

	explanations and explorations were a-priori predictions and decisions.” This is both a strong claim and presumably within a framework that somehow a priori hypotheses are different to a posteriori ones in some deep way (e.g., see Kataria, 2016). This could benefit from being unpacked and explained to the modellers being addressed, or indeed any readers. Especially so since many in mathematical psychology are Bayesians in multiple senses.	(this is a separate empirical question), or (2) that the distinction between a-priori predictions/decisions and post-hoc predictions/decisions is meaningful in all circumstances (this is a separate philosophical, as well as potentially empirical, question). As discussed earlier, our goal here is to provide a concrete proposal of a preregistration template that advocates can extend and critics can critique, rather than fundamentally argue the merits of preregistration. However, we agree that it is important to note that there is a broader philosophical debate related to this issue, and we have added a footnote on this on p.17 (as well as the very useful reference to empirical evidence on this matter which you have suggested here).
Reviewer 1, Main Body, Comment 17	On page 17, if you already have the code how are you preregistering? How can you check your code works if not by already running it against previous data, to have module testing, etc.? The authors here need to deeply engage with the point in my paper they cite, (Cooper & Guest, 2014). This paper explains how code is not sufficient to constrain things in a meaningful way without a specification.	Good point; we agree that this important point was unclear in the previous version of our manuscript. As we state in the text, such prespecification via code would have to be substantial and include extensive documentation. Furthermore, we have added a sentence to add a point related to your 2014 paper, which emphasises just how extensive this additional documenting work would have to be: “Furthermore, the documentation would have to be much more extensive than standard code documentation, as it would also need to include information on the theory behind the model to enable meaningful interpretation (Cooper & Guest, 2014).” (p.20)
Reviewer 1, Main Body, Comment 18	On page 18, “Rather, we believe that our preregistration template may be a useful tool to help ensure robustness and transparency in model application studies”. Is this also not (at least ultimately) an empirical claim? I’m not entirely sure what robustness is in this context, and perhaps it might	We see the reviewer’s point here, though as we highlighted elsewhere in the paper, the goal of this manuscript is to propose a template, not to evaluate it -- as this will not be possible until other researchers use it. Therefore, we have clarified that in this sentence as well: “Rather, we believe that our preregistration template may be a useful tool to help ensure robustness and transparency in model application

	benefit from description or definition?	studies; whether this is the case is ultimately an empirical question. " (p.21)
Reviewer 1, Main Body, Comment 19	On page 19, "Any post-hoc addition or modulation is then clearly exploratory rather than confirmatory research." What if the modellers are using an existing model but while writing the preregistration did not realize they misunderstood something in the original model? This is extremely common in modelling, especially when re-implementing models (Cooper & Guest, 2014; Guest et al., 2020).	We agree that this sentence was probably written too strongly, and we have revised it to be more nuanced: "Any post-hoc addition or modulation is then considered to be exploratory rather than confirmatory, unless the changes can be justified as being due to technical mistakes in the original preregistration document (e.g., the researchers state that they wish to implement a specific model, but then incorrectly specify the model within the preregistration, see Cooper & Guest, 2014; Guest et al., 2020 for a discussion). However, in the case of technical mistakes in the original document, we believe that the best practice would be for researchers to create an updated version of the preregistration document when they realize this mistake, as well as adding any new experience that they may have with the data since the original document was written (as in 12-14 of the preregistration template)." (p.26/27)
Reviewer 1, Main Body, Comment 20	On page 25, in fact this whole section on a computational model and a specific experimental setup, I am not familiar with. So my comments here must be taken with a pinch of salt. As with all modelling, in my experience, it's very hard to have an informed opinion without some playing around with the implementation, if not building it from scratch. On page 25, "The assessment in Evans & Brown (2017) was rather qualitative and not very rigorously defined, making this a good opportunity to show how preregistration in cognitive	While we see the reviewer's point here, we did not intend to make the claim that our specifications in our example preregistration were significantly more rigorous/transparent than it would be possible for any researcher to do without our preregistration template, but rather that we used the template to try and improve these aspects from the original Evans & Brown (2017) study. Therefore, we do not believe that the suggested contrast condition would be effective in helping any of our claims. However, we agree that this may have been unclear in the previous version of our manuscript, and have added the following text to our revised manuscript: "Note that we are not intending to claim that our preregistration template enabled us to provide a more rigorous or transparent assessment than any researcher(s) could have possibly

	modelling can add rigor and transparency in situations with many potential researcher degrees of freedom.” As we know from modelling and theorizing, such comparisons can easily be spun, and often are, without controls in place, to favour a specific outcome. Hence, why might it be good to consider a contrast condition without preregistration perhaps?	managed without using our template, but rather showcase how our use of the template allowed us to add (what we believe to be) more rigor and transparency to the assessment.” (p.29/30)
Reviewer 1, Main Body, Comment 21	On pages 27-28, why do the authors choose to define the hypotheses qualitatively? I don't know what is more appropriate, but a formal definition with reference to the expected effects might be possible too here and useful.	We would like to clarify that we do not define the hypotheses qualitatively, but define one (of three) of our tests of the hypotheses as a qualitative one. We describe this as being qualitative as it does not involve formal statistical inference, and instead uses parameter estimation and posterior predictives (similar to approaches advocated by some of those who fall in the “parameter estimation” camp of inference, such as John Kruschke). While it is theoretically possible to define these assessments more formally (some of which we do in the other tests of the hypotheses, and others are too computationally intensive to be feasible), we believe that the additional qualitative assessments help to provide a clearer picture of how people differ from optimality.
Reviewer 1, Main Body, Comment 22	On page 31, “Nevertheless, deviations from a preregistration should always be possible, for example if the researcher gained important knowledge in between writing the preregistration and analysing the data”. Doesn't this always happen with modelling, and so the modeller will always be forced to deviate? This might not be a problem, but needs to be discussed.	While we see the reviewer's point, we would argue that saying this “always” happens is a strong claim, and one that is not necessarily true (particularly in the case of model application, which is the area of cognitive modelling that our template is intended for). For instance, if the researcher already has the model coded (e.g., if it's a model that they have applied before to other data, or they have applied the model to simulated data for a parameter recovery study), and the data is in an “easy-to-parse” format, they may start analysing the data immediately after posting their preregistration document. Therefore, we do not think that discussing this point is

		within the scope of the current paper, as it is currently empirically unclear how often researchers using our template would gain new knowledge between posting their preregistration document and beginning to analyse their data.
Reviewer 1, Main Body, Comment 23	On page 36, “No preregistration is perfect, and using an overly general template – or one designed for a different field of research – is likely better than using no template or eschewing preregistration entirely.” What does better mean or imply here? Either way this needs unpacking because it’s a very strong claim and does not seem superficially true (if at all).	Good point; we did not intend to make a strong claim here, and have changed “is likely better” to “may help to provide more information”.
Reviewer 1, Final Comments	Overall I think this paper and concept need a lot more work to actually be appealing in a meaningful way to the communities that seem to be addressed. Even though I remain unconvinced of the scientific value of preregistration here, clarifying all the points I mentioned above are likely to heighten the contradictions (for people like me who disagree) but also likely to further engage whoever the intended audience is (it is not fully clear to me which exact communities are being addressed). Ultimately, it will fall to modellers and their communities if they wish to adopt a checklist. In my opinion, such checklists, if not couched and framed properly, can easily debase and devalue modelling work making it inherently unscientific by unhinging models from the context via a mindless box	Thank you for critically engaging with our manuscript! We understand that you have a different view to us regarding the utility of preregistration, and appreciate that despite this you still took the time to leave us such detailed feedback, which we have attempted to address throughout our revised manuscript. As discussed in our response to Reviewer 1, General Comment 1, our intention was to direct our manuscript at *both* the modelling and open science communities, and we hope that our revised manuscript better achieves this goal. Furthermore, as noted in our response to Reviewer 1, Abstract, Comment 4, our preregistration template is only intended for a specific type of preregistration -- model application -- and not for the more exploratory types of modelling, such as model evaluation or model development. Moreover, as discussed in our response to Reviewer 1, General Comment 3, we certainly do not wish to force modellers who disagree with preregistration to use preregistration, and in terms of the modelling community, our template is directed at those who are either on the fence about preregistration for model application studies, or who wish to

	ticking exercise. This is likely possible to be tempered, controlled for, and even avoided, I think, if the authors try to be clearer with their ideas and prose. I hope my comments cause pause for thought and further exploration of their own ideas and proposals which seem to be prescriptively stated, again something that can be side-stepped with more careful language and perhaps deeper conceptual analysis and/or engineering.	preregister their model application studies but are unsure how to do so. To us, preregistration is an optional step that is the choice of the researcher.
Reviewer 2 (Alex Holcombe)	This is a useful contribution to the literature that I think will improve work that uses cognitive and related models to estimate parameters that purport to measure mental constructs. The manuscript provides a novel preregistration template, which appears to be the first of its kind, applying a cognitive/mathematical model to behavioural data to estimate the model's parameters. First, I should say that I have done very little cognitive modeling myself, so the expertise I bring to this review is more limited to various aspects of open science, preregistration, and experience with studies that focus on within-participant estimation with many trials per participant. I think this is a needed contribution to the literature and I	Thank you for your review and for your positive evaluation of the contribution of this paper to the literature.

	think the field will be appreciative of this work. My specific points about the manuscript, below, are largely about presentation and a few points that I think were missed.	
Reviewer 2, Comment 1	“the added transparency of openly stating pre-existing knowledge will hopefully provide similar credibility, and at the very least enable the reader to put the results into context (Nosek et al. 2018).” I don’t think that the preregistration skeptics will appreciate that phrase much, because shouldn’t the Introduction to the resulting paper put the results into context anyway? Or maybe the authors mean something else, if so it could be clarified.	Good point; we agree that our previous wording was somewhat ambiguous, and we have attempted to clarify this in our revised manuscript. Our intention was to make the point that the preregistration of secondary data sets can provide transparency surrounding the context of how the research process occurred that led to the results (i.e., what they knew about the data before proposing their planned analyses). We hope the following change removes that ambiguity (on page 15): “... and at the very least provide the reader with more context on the research process that produced the results presented in the corresponding paper (Nosek et al., 2018).” (p.17)
Reviewer 2, Comment 2	As the authors are aware (they indicate it with their citation of a Morey 2018 tweet), the overall value of preregistration for cognitive modeling is presently controversial, for example one paper was titled by its authors “Preregistration is redundant, at best”. For the purposes of this review, I will refer to such opinions as those of “preregistration skeptics”. I think the recent rise in preregistration skeptics makes it particularly important to ensure the writing about preregistration is clear and judicious. I think that it is NOT the job of this paper to settle those debates or even to take them head on. Clearly many researchers find preregistration	Thank you for this comment. This was the exact scope and context that our manuscript intended to achieve (i.e., to make a concrete proposal of a preregistration template for model application for those who are interested in preregistration, rather than to debate the merits for preregistration), and we hope that our revised manuscript achieves our goal.

	valuable, making the provision of a template valuable. Because this paper is about estimating parameters with a model, some of the preregistration skeptics' concerns are not applicable, and the authors point that out, which is useful. But here are a few comments:	
Reviewer 2, Comment 3	The authors describe the purpose/benefits of preregistration variously, and these different benefits are scattered through the manuscript. It might be a good idea to bring them together. Appropriately, pretty early in the Introduction the researchers provide an extended motivation for preregistration including that the purpose is "to help researchers avoid fooling themselves into believing that their post-hoc explanations and explorations were a priori predictions and predictions." but at a later section they add that "preregistration seems to be useful both for the a-priori justification of choices, and for the clarification of which choices do not have clear justifications."	Thank you for this comment -- we have now included all the benefits of preregistration that we mention throughout the paper in the "Preregistration" part of the introduction. We think that it is sensible to still mention specific advantages of preregistration where it is relevant to do so later in the paper, and we hope that you agree.
Reviewer 2, Comment 4	Some preregistration skeptics suggest that what is pre-planned is irrelevant to the inferences made after the data are in, because the theory is the theory regardless of what the researchers thought or didn't think about its predictions before the study was done. What this misses is that even putting aside	Thank you for adding this very valuable contribution that preregistrations make to the research landscape. We have added this to our list of potential benefits of preregistration, and we have added a footnote thanking you as we were unable to find this point made in the literature:

publication bias and selective reporting, and even in those very few circumstances in which a theory is fully specified, enough for a second researcher to determine exactly how diagnostic the experiment was for adjusting their prior on that theory, fully evaluating that (how much to adjust one's prior and working out how strongly the theory predicted or didn't that result) is a huge amount of work, and the expertise required to do it is in very short supply. For many theories, there may be only a few dozen people in the world qualified to do it. Therefore one of the great benefits of preregistration is that it provides a credible signal of accurate prediction which has much lower cost to evaluate. That is, if a reader sees that another set of researchers made several preregistered predictions and those turned out to be correct, that's already a pretty good signal for the reader to increase the credibility of the theory that the researchers said they were using to make the prediction. Now, to be more confident in that, the reader should make some evaluation of whether it looks like the theory really did make those predictions, and that other theories didn't, which is the work that needs to be done without preregistration to *even get started* knowing whether to increase or decrease one's credence in the theory. Preregistration provides a shortcut, giving readers a rough sense already, even if it will be somewhat unreliable, it is a lot better than nothing. In a world with finite expert peer resources, we do benefit from this. I imagine some paper on preregistration in the literature already points out this benefit, although in my limited experience the	"This is not only helpful for the researchers themselves, but may also serve as a basic but credible signal of accurate prediction for readers not familiar with a field or theory, which may be particularly useful for niche and intricate theories and areas of research." (p.4/5)
--	--

	preregistration skeptics seem to ignore it.	
Reviewer 2, Comment 5	I noticed that the word “multiverse” doesn’t appear in the manuscript, but the authors probably know that this is roughly a term for the phenomenon that the authors refer to as robustness checks, so they may want to mention this term there as well.	Good point; we agree that it is useful to mention the term “multiverse” as well. Within mathematical psychology, the idea of robustness checks existed some time before the multiverse analysis paper, and therefore, we wish to keep the primary term as one that researchers in mathematical psychology are most familiar with. However, we agree that linking robustness checks to the exact term “multiverse analysis” will be useful for some readers, and therefore, we have made the following change in the revised manuscript: “One previous argument against the utility of preregistration has been that cognitive modelling research has other, superior practices for ensuring the robustness of their findings, such as robustness checks (see e.g. a tweet by Morey (2018); robustness checks are similar to what has recently been referred to as ”multiverse analyses” in the context of psychological research (Steege et al., 2016).” (p.19)
Reviewer 2, Comment 6	I really like the inclusion in the template of issues and contingency plans!	That’s great to hear, thank you! Including these was a very helpful suggestion by a previous reviewer (Jeffrey Starns), who we now also credit for the idea by name.
Reviewer 2, Minor Comment 1	Up to page 6, aspects of the introduction are somewhat redundant, so I think the manuscript would have more impact if the authors tried to address this.	Thank you for pointing this out. We hope we have addressed this worry sufficiently.
Reviewer 2, Minor Comment 2	p.21 “If you are not interested in the parameters and are going straight to statistical inference without estimating the parameters” I wasn’t sure what was meant by this, so I think many readers may have the	Good point; this was unclear in the previous version of the manuscript, and in the revised manuscript we have attempted to clarify what we mean here: “If you are not interested in the parameter estimates and are purely focused on statistical inference about

	same problem; probably it means if you just want to do something like a statistical test to assess whether the parameter values are higher in one condition or another - that might be worth clarifying.	differences between conditions/groups using a method that does not directly require initial parameter estimation, please state this clearly and motivate this choice.” (p.24)
Reviewer 2, Minor Comment 3	About outlier exclusion, the manuscript suggests that “In purely experimental studies, this is usually focused at the level of entire participants. “ I think this is overstating things - in my experience, experimental studies frequently have outlier exclusion, for example trials with response times greater than a certain number, or greater than a certain number of standard deviations than the mean.	Thank you for pointing this out. We have replaced “usually” with “often” in this sentence.
Reviewer 2, Minor Comment 4	page 15, It’s weird to start a new section with “Moreover” and it’s not clear what the word means here.	Thank you for pointing this out. We agree and have simply removed the word “Moreover” here.
Reviewer 2, Minor Comment 5	Page 26: the word is “kinematogram”, not “kinetogram”. Incidentally, neither word appears in the paper cited.	Thank you for pointing this out! We have changed this to “random dot motion task”
Reviewer 2, Minor Comment 6	“One potential critique of our preregistration template could be that the prompts are too open-ended. However, it has previously been found that a format with specific, open-ended questions is better at restricting researcher degrees of freedom than a purely open ended template (Veldkamp et al., 2018). “ This doesn’t seem to	Thank you for pointing this out. This was indeed not as clear as it could be. We have clarified this as follows: “One potential critique of our preregistration template could be that the prompts are too open-ended. However, while the questions are open-ended, they are asked in a structured format. It has previously been found that a structured preregistration template format with

	make sense. From the structure of the sentence, I was expecting the last thing, “open ended template” to be a contrast with “specific, open-ended questions”, but they both say they are open ended and I don’t understand what the difference is.	specific but open-ended questions is better at restricting researcher degrees of freedom than an unstructured, open-ended template (Bakker et al., 2020).” (p.37)
Reviewer 2, Minor Comment 7	For the plate diagrams, it would be helpful to indicate what the subscripts, e.g., i and j, are iterating over.	Good point; previously we only indicated this in the appendix (i indexes participants, and j indexes blocks). However, we agree that this may be difficult for the reader, and have added clear labelling to the diagrams in the paper.
Reviewer 2, Minor Comment 8	For one of the plate diagrams, the caption includes “comparing the posterior distributions of the decision threshold parameters against the posterior predictive distributions for the optimal threshold”, in other words that two things are compared, but it’s not clear what is which. E.g., what shows the posterior predictive distributions for the optimal threshold? Is it the green region? In any case, please indicate what the green stuff is in the caption.	Thank you for bringing this up; our previous figure caption was quite unclear, and we have adjusted in the revised manuscript to the following: “Plots comparing the posterior distributions of the decision threshold parameters (dots with error bars, reflecting the posterior median and the 95% credible interval, respectively) against the posterior predictive distributions for the optimal threshold (green area; lightest shade being the 0.4 to 0.6 quantile region, the middle shade being the 0.2 to 0.4 and 0.6 to 0.8 quantile regions, and the darkest shade being the tails of the distribution), for the fixed time group (left) and the fixed trial group (right).” (p.34)
Reviewer 3	This is an excellent paper about preregistration cognitive model application. The authors discuss the specific researcher degrees of freedom for this type of research and make a clear case why we need this additional preregistration template. They are also very clear for which type of cognitive models it is and isn’t a good template.	Thank you for your review and for your positive evaluation of the contribution of this paper to the literature.

Reviewer 3, Comment 1	On pages 11 and 12, the additional degrees of freedom are mentioned, and on pages 19 and following, the new sections for the preregistration template. I got a bit confused by the different letters that are used (M, E, I, and RC for the RDFs, and A, B, C, and D for the sections). They seem to overlap partly, but not completely. I think that it will be helpful to relate the new RDFs to the specific new sections. In this way, it is clearer how these sections will prevent the opportunistic use of these additional RDFs.	Good point; we agree that the previous format may have caused some confusion, particularly as A-D are sections, whereas E is a RDF. We have attempted to make the link between the new RDFs and the specific new sections clearer by adding the following paragraph on page 22 (under “Preregistration Template”): “The sections below correspond to most of the modeller's degrees of freedom proposed above. Specifically, section A corresponds to the modeller's degrees of freedom M1, M2, and M3; section B includes questions aiming at E1, E2, E3; and section C asks about robustness checks (i.e. RC1). The modeller's degrees of freedom I1 and I2, related to statistical inference, are covered in slightly amended existing preregistration template items.”
Reviewer 3, Comment 2	I wasn't sure of the function of the example application (starting at page 24). On page 6, the authors mention that it is to “showcase how it can help to constrain researcher degrees of freedom”. Therefore, I expected an example preregistration and a discussion of the RDFs that are prevented. But it is more a general research example (including results and discussion) with only limited information about the preregistration itself. The preregistration itself is put in the appendix. It would be helpful if the authors clarify why they included an example application (the goals) and add more of the preregistration in this example preregistration.	This is an excellent point, which is similar to Reviewer 4, Comment 12. We agree that the purpose of our example application was somewhat unclear in the previous version of the manuscript, and we have attempted to clarify this in our revision. Specifically, the goals of our example application were to (1) show that it is feasible to use our proposed preregistration template in a realistic model application context, and (2) provide an example (though not necessarily a perfect one, as discussed in our response to reviewer 4) on how to use the template. We have attempted to further clarify this by adding the following sentence to the beginning of our example application (page 28): “We include an example application of our template to showcase that using our proposed preregistration template for model application is feasible, and provide an (imperfect) example of what a preregistered model application using our template might look like.”
Reviewer 3, Additional Comment 1	In the second paragraph of page 3, preregistration is defined and explained. I miss that	Thank you for alerting us to this omission -- we have corrected this now: “Preregistration involves the

	preregistrations should be time-stamped/frozen registrations.	specification of a researcher's plans for a study, including hypotheses and analyses, typically before the study is conducted. This usually takes the form of a time-stamped document that contains these plans, which is made available online. " (p.4)
Reviewer 3, Additional Comment 2	In the same paragraph, the different reasons to preregister are mentioned. However, I miss the transparency reason, although transparency is mentioned as an important reason for preregistration in the remainder of the paper (e.g., p. 8).	Added: "It has been suggested that preregistration can help to make the research process more transparent, (...)" (p.4)
Reviewer 3, Additional Comment 3	The study by Veldkamp et al. is now published: Bakker, M., Veldkamp, C. L., van Assen, M. A., Cromptvoets, E. A., Ong, H. H., Nosek, B. A., ... & Wicherts, J. M. (2020). Ensuring the quality and specificity of preregistrations. PLoS Biology, 18(12), e3000937. https://doi.org/10.1371/journal.pbio.3000937	Corrected, thank you.
Reviewer 3, Additional Comment 4	On page 4, row 12, I would add the PNAS paper by Nosek et al. because they use postdiction and prediction in that paper.	Done, thanks!
Reviewer 3, Additional Comment 5	On page 14, secondary data is discussed. An essential aspect of preregistering secondary data is discussing prior knowledge of the data (this is also part of the model application template). This helps authors to be transparent about what they know and do not know of the data set and allows readers to evaluate this. I think that it is good to mention this aspect of registering secondary data as well.	Good point; we agree that this point could have been clearer in the previous version of the manuscript. In the revised version of the manuscript, we now state (on page 17): "Although such a preregistration will not function in quite the same way as it is meant to with original data, the additional transparency of openly stating all pre-existing knowledge of the specific secondary data sets that one plans to analyse will hopefully provide similar added credibility, and at the very least provide the reader with more context on

		the research process that produced the results presented in the corresponding paper (Nosek et al., 2018).”
Reviewer 3, Additional Comment 6	On page 32, changes after preregistration are discussed, and it is stated that these should be motivated and reported in a transparent manner (row 24). Can you give some examples or references on how to report this transparently? I think that this is very helpful because this is something that currently often goes wrong, as you discuss here as well.	Good point; we have now added a specific reference that discusses and provides a few examples of transparently reported and (in some cases) motivated changes after preregistration. “Moreover, minor accidental omissions which likely do not affect the outcome can happen with any preregistration, and should be transparently reported (for a discussion and a few examples of how to transparently report and justify deviations, see Claesen et al. 2019)” (p.36)
Reviewer 4 (Wolf Vanpaemel), Comment 1	The authors provide a preregistration template for situations in which data analyses require the application of a cognitive model rather than an off-the-shelf model like e.g. a regression model. Quite bluntly, I am not convinced of the usefulness of a specific template for this situation. My hesitation does not stem from the fact that I don't believe in the usefulness of preregistration when engaging in cognitive modeling (in fact, I have proposed a format for a Registered Report specifically geared towards theory testing, designed to assure severe tests of cognitive models; see Vanpaemel, 2019). Rather, I am skeptical because the current template overlaps quite extensively with the standard OSF checklist (https://docs.google.com/document/d/1DaNmJEtBy04bq115OxS4JAscdZEKUGATURWwnBKLYxk/edit?pli=1#). I realize this is very much my personal opinion, but I	Thank you for your detailed and helpful review. We appreciate your expertise as a modeller and the insight that you have provided, and we hope we have appropriately addressed your questions and comments in our revised manuscript. Regarding this first major concern, we believe that it is somewhat misleading to say that our template is redundant simply because it overlaps with the standard OSF checklist. Within our manuscript, we are very clear that our template is an *extension* of the existing OSF checklist -- as many of the other more specific preregistration templates and checklists have been (e.g., Flannery, 2018; Haven & Grootel, 2019; Paul et al., 2021; Kirtley et al., 2021). Furthermore, our manuscript only focuses on our extensions of the standard template for use in model application (i.e., in our *manuscript* we are not repeating content from the standard OSF checklist). While we see the reviewer's point that our template is not completely unique from all other templates, we believe that questioning the value of our template for

	doubt the few deviations warrant a checklist of its own.	this reason would be similar to questioning the value of a new model because it extends from an existing model (e.g., questioning the value of Ratcliff's 1978 diffusion model -- one of the most influential models in mathematical psychology -- because it only adds a single parameter to Stone's 1960 diffusion model). Instead, we believe that templates should be assessed on *how much value the extensions add* (e.g., similar to how model extensions are assessed) for those who wish to preregister model application studies, but may not be sure how to do so. Concerning the usefulness of our template, and more generally, the usefulness of specific templates that extend from the existing standard OSF checklist, we believe that there are several reasons that they are desirable, including:  1) Specific templates make the application of preregistration more straightforward for researchers; They don't have to reinvent the wheel every time, and they can build on something that other researchers in their area have used and hopefully improved on before. 2) Templates are helpful for standardisation. If every modeller wanting to preregister a model application study tries to use the standard OSF checklist for the purpose of model application, we will have many different versions trying to do the same thing, but probably doing it in very different ways. Our proposed template would at least give one common starting point. 3) Specific templates may encourage researchers to be exhaustive in their preregistrations. Using a "standard" preregistration template, you run the risk of missing things out. 4) As argued by Reviewer 2 Comment 4: a preregistration can serve as a basic but credible signal of accurate prediction
--	---	--

		for readers not familiar with a field or theory, which may be particularly useful for niche and intricate theories and areas of research, such as much of cognitive modelling. Therefore, lowering the threshold for preregistration by proposing a specific template for model application in cognitive modelling seems worthwhile and important. 5) Most importantly: Given the contentious debate surrounding preregistration in this particular area, a template for a part of cognitive modelling also represents a counter-argument to the idea that preregistration in cognitive modelling is simply infeasible in all areas and categories of cognitive modelling. 6) Finally, the preprint of this manuscript has been cited 7 times (including Corker, 2021) and has been downloaded more than 500 times. This indicates a general interest in this type of template in the broader research community. We are also aware of at least one preregistration that uses the template: https://osf.io/7gt45/. Corker, K. S. (2021). An Open Science Workflow for More Credible, Rigorous Research. Overall, we believe the question of redundancy/importance is a question of A) are specific templates potentially useful?; and B) does our specific template provide additional material that may assist researchers in preregistering model application studies (at least to a similar extent as other previous extensions)? We believe that A) is the case, and that our template achieves B) to at least a similar extent as previous extensions, and we hope that our responses and revisions also convince the reviewer of this. Having said that, we agree that Vanpaemel (2019) presents an interesting discussion of the pre-registration of a model evaluation study, and we now cite this paper in our
--	--	---

		manuscript (p.39).
Reviewer 4, Comment 2	As far as preregistration is concerned, I don't see a fundamental difference between registering the use of, say, a SEM type model and a cognitive model. Both require the exact specification of the model version (including which parameters to include) and the estimation technique (say OLS vs multilevel approach). This intuition is further strengthened by the fact that the distinction between cognitive and statistical models is sometimes moot: MDS started as a cognitive one, but is now treated as a statistical one; the cognitive model FLMP turned out to be equivalent to the statistical Rasch model, and so on. That, together, with a desire for parsimony, makes me doubt the usefulness of this template.	While we see the reviewer's point, we believe that answering the continuing philosophical modelling question of "where is the border between statistical and cognitive models?" is far beyond the scope of the current manuscript. We agree that statistical models and cognitive models can share many similarities (and that researchers often debate over which models are "statistical" and which are "cognitive"), but opinions amongst experts are very much split as to whether statistical and cognitive models are (1) distinct, (2) different ends on a continuum, or (3) the same thing. We believe that the reviewer makes good arguments for opinion 3 in their review, but again, the aim of our paper is not to debate the statistical/cognitive model distinction. At a practical level, we do agree that there are similarities between "applying SEM to a data set" and "applying a cognitive model to a data set" (and therefore, perhaps our template could be of some use for SEM, though we would not directly recommend it for SEM researchers), but we do not believe that these processes are identical. For example, while both involve the specification of a model, model application involves using one (or more) of many potential models within one of many potential classes models, where the exact relationship between the latent constructs is defined by the theory. In contrast, the options for "classes" of models in SEM are arguably narrower, and the exact relationship between the latent constructs is often part of the estimation process. Furthermore, while SEM is often focused on estimating the relationship between latent constructs, model application is focused on assessing how latent constructs (i.e., the model parameters) differ between conditions/groups. Again, we are not attempting to claim that applying SEM and applying cognitive models are without similarities, but we think that there are sufficient practical differences

		for one to at least acknowledge that the process for applying them can greatly differ. Furthermore, we disagree with the point that a “desire for parsimony” makes our template less useful. We agree that having an extremely large number of templates could potentially make the landscape of templates difficult for researchers to navigate, e.g., if there were a template for applying the diffusion model to research designs with a single manipulation that has two levels, as well as a template for applying the diffusion model to research designs with a single manipulation that has three levels. However, we are not at the stage where there are too many existing templates; this is the first template for any category of cognitive modelling. Furthermore, we believe that having and creating more resources for researchers who wish to perform preregistration is useful (see our response to the editor and to Reviewer 4, Comment 1).
Reviewer 4, Comment 3	Personally, I would much rather read a very brief paper with some caveats and good practices when using an existing pre-reg template when a cognitive model is being used, rather than having yet another template to choose from. Nevertheless, I am not to decide on the usefulness, so I will provide some comments for improvement, assuming others find the new template more useful than I do.	While we see the reviewer’s point, we believe that there are already several papers that talk about good practices in modelling that relate to preregistration (e.g., the Computational Brain & Behavior special issue from 2019, led by the target article of Lee et al.), as well as several blogs posts (linked below) that broadly discuss these issues. However, we believe that one thing that is lacking from the literature is a specific, concrete proposal on how researchers could go about preregistering some types of cognitive modelling research. As mentioned by Reviewer 2 (Comment 2), our goal was to provide a specific, concrete template for preregistration in model application, which we hope will assist researchers in preregistering model application studies (if they wish to do so), and also give proponents and critics a concrete template to extend upon and/or directly critique. References:

		Lee, M. D., Criss, A. H., Devezer, B., Donkin, C., Etz, A., Leite, F. P., ... & Vandekerckhove, J. (2019). Robust modeling in cognitive science. Computational Brain & Behavior, 2(3), 141-153. https://featuredcontent.psychonomic.org/arguments-for-preregistering-psychology-research/ https://featuredcontent.psychonomic.org/prediction-pre-specification-and-transparency/ https://www.bayesianspectacles.org/dont-interfere-with-my-art-on-the-disputed-role-of-preregistration-in-exploratory-model-building/
Reviewer 4, Comment 4	It is unlikely that this version of the template is the ideal or the final one (or even if it is ideal, will stay ideal in a continuously moving landscape). Therefore, I would encourage the authors to actively seek feedback from users of the template, and adapt the template continuously based on the feedback (with a proper versioning system).	We completely agree with the reviewer here! Our hope is that we, as well as others, will alter and extend the current template in the future, and we think that this would be an important part of any potential progress in the preregistration of model application (and perhaps other types of modelling in the future). For this reason, we have already added our template to the OSF, and we are hoping that it will be continuously adapted by us and/or others in the future. In fact, our template was already used as the basis of a group discussion on preregistration in cognitive modelling at a workshop held by the centre for open science in early 2020.
Reviewer 4, Comment 5	I haven't read the papers introducing the four types of cognitive modeling, but I was wondering why this example wasn't a case of model comparison (clearly, using the Bayes factor, two models are being compared, so surely I am missing something).	Good point! A similar issue was also brought up by Reviewer 1 (Main Body, Comment 12), so we have attempted to clarify some of these distinctions. Our definition of "model comparison" (and how it differs from "model application", even though both can often use model selection methods, such as Bayes factors) is that it involves the quantitative comparison of theoretically distinct models (i.e., models with at least

		some difference in the actual underlying process proposed), rather than using a single model to determine whether or not parameters differ between conditions/groups. We have clarified this when we mention the concept of “model comparison” on page 10, which now also states: “Note that while model application and model comparison can both make use of model selection methods, they are separated by their intended purpose. While model application aims to determine whether parameters differ between conditions/groups within a model, model comparison aims to quantitatively compare theoretically distinct models, where the models differ in at least some aspect of the proposed underlying process.”
Reviewer 4, Comment 6	Not everybody agrees that the distinction between exploratory and confirmatory analyses is meaningful and/or useful. Maybe this position should be acknowledged or discussed (see Szollosi and Donkin, 2019).	Good point; this is something that we should have mentioned in the previous version of the manuscript. In our revised manuscript, we now explicitly state: “This is important as each of these practices can render inference based on seemingly confirmatory analyses unwarranted (Wagenmakers et al., 2012; though see also Szollosi & Donkin, 2019 for a discussion on the potentially problematic dichotomy between confirmatory and exploratory analyses)” (p.4) We have also added a footnote after this clarification to unpack the issue a bit further: “More specifically, in discussions surrounding preregistration, research is often dichotomised into strictly exploratory and strictly confirmatory research. “Confirmatory” research refers to research where hypotheses and analyses were planned before the start of the study. However, this dichotomy is not necessarily a useful one (Scheel et al., 2020, Szollosi & Donkin, 2019), and exploratory and confirmatory research are likely part of a broader spectrum. Nevertheless, we will use these terms in

		the context of describing the debate surrounding preregistration in cognitive modelling to reflect the language and concepts used by both proponents and critics." (p.4)
Reviewer 4, Comment 7	I am not sure whether the authors simply borrow terminology from Veldkamp, or speak with their own voice, but I would be very hesitant to use the term "the quality of preregistrations". Veldkamp et al. have looked at scope and level of detail, which are (fairly) objective characteristics, but equating these with quality is a strong epistemological/meta-scientific commitment.	Good point, this was poor wording on our behalf. We have now changed this to "specificity".
Reviewer 4, Comment 8	I think "E2: Specifying settings/priors for parameter estimation." is misguided. As I have extensively argued elsewhere, especially in cognitive modeling, priors should be seen as part of the model (in some sense, models *are* predictions, and without priors, we can not make predictions; Vanpaemel, 2010; Vanpaemel and Lee, 2012). This is not to say that priors should not be part of the preregistration plan, but I think they are more fit in the M category. (The same holds for B.3) The authors are of course free to disagree with me on the role of the prior in cognitive modeling, but I would like to make sure that putting priors in E and not in M is a deliberate decision on their part.	We see the reviewer's point here, and agree that the priors can certainly be considered part of the model, instead of part of the estimation method. Our reason for placing them in the estimation section is that some cognitive modelling researchers do not use (or even agree with) Bayesian methods. Some researchers very strongly believe that even when priors are used, they should be considered a part of the method, rather than a part of the model itself that is not inherently Bayesian (e.g., personal communications with Thomas Palmeri and Gordon Logan). Therefore, we believe that placing the priors in the estimation section makes the most sense given the potential scope of the audience. However, we agree that there are strong arguments for why the prior should be considered part of the model, and we now explicitly note this within the revised manuscript: "However, note that there are arguments for why the priors should be considered as part of the model itself, rather than part of the estimation method (Vanpaemel, 2010; Vanpaemel, 2012). Our choice to place the priors in the estimation section is not intended as a

		stance on this theoretical issue, but rather to make our template as applicable to both Bayesians and frequentists as possible.” (p.13)
Reviewer 4, Comment 9	I think it is a dangerous practice to interpret or do tests on parameters estimated using models that might not be appropriate for the data at hand. It seems that at least a minimal check of descriptive adequacy (using e.g., posterior predictive checks) should be included, or more broadly, the conditions under which one is confident enough in the model to work with the estimates. (To be fair, this comment is not restricted to cognitive modeling, and descriptive adequacy should also be checked when e.g regression is used.)	While we generally agree with the reviewer’s point regarding descriptive adequacy, we believe that adequacy assessments constitute model evaluation (i.e., assessing how well a model can account for specific trends in the data), and therefore, is separate from the process of model application. Therefore, we believe that while these assessments are still important, they should not be covered by a template dedicated to model application, and instead should be covered by a potential template dedicated to model evaluation, to ensure that these assessments of descriptive adequacy are rigorously specified. These are issues that we discussed within the limitations section of the previous version of our manuscript, though we agree that this may be too late to bring attention to these important points. Therefore, we now mention and refer readers to this point on page 10: “Furthermore, it should be noted that assessing the descriptive adequacy of a model -- i.e., performing model evaluation to ensure that the model provides a good account of the trends in the data -- is also often a part of studies that implement model application (see the “Limitations” section for a more detailed discussion).”
Reviewer 4, Comment 10	I (for one, but that is again just my personal opinion) strongly disagree with the necessity of words such as "only" in pre-reg plans. It makes pre-reg plans overly legalistic, and, most importantly, these words seem to be implied, by Gricean maxims. If my son asks me what he can choose for dessert, and I would respond "There is cake", he would be surprised (and angry/or annoyed with my dad joke) when	While we agree with the reviewer’s point that the word “only” may not be necessary in all situations, we believe that placing this word as a qualifier can easily serve to reduce ambiguity in the research plans, as it does not leave the interpretation up to unstable implications and assumptions. Take a further cake example that is analogous to your theta/delta case (i.e. “and” not “or”): if we said “there is cake” and were in the UK (i.e., these assumptions likely depend on our cultural backgrounds),

	he would find out there is not just cake but pie as well, even though what I said is not a lie and technically correct. He is just working on the assumption that if I answer his question, I am being exhaustive (unless I use words like “for example”). I believe something similar happens when someone writes in a pre-reg plan "we will vary parameter theta". If these authors then end up varying both theta and delta in their paper, this is technically correct (i.e, consistent with the pre-reg plan), but I think most readers will be surprised to see this, as they will work on the assumption that "we will vary parameter theta" is exhaustive.	you might expect that there would also be tea. However, specifying that “there is only cake” would make it very clear that there is no tea involved, and may change your decision on whether to accept the invitation. In the everyday context, not making such assumptions explicit is often fine. However, we believe that in a research context, not making such assumptions explicit can be potentially problematic. It is conceivable that "we will vary parameter theta" is not clearly exhaustive in all research contexts and to all researchers, and adding the word “only” helps to reduce potential ambiguity. Furthermore, we believe that including the word “only” has a very low cost associated with it, so if one is already going to the trouble to write the preregistration, then there seems to be little reason not to add the additional word to remove any potential sources of ambiguity. However, we agree that the necessity of the word “only” is certainly a point of potential debate, and we now explicitly mention this within the revised manuscript by adjusting the text to state that “It should be noted that in order for a preregistration to be exhaustive, we believe that it is important to repeatedly use clarifying words such as “only”” (p.23), and adding a footnote to further clarify the point: “While it can be argued that the word “only” is implied in a preregistration, as the preregistered analyses are the only ones mentioned within the preregistration, we believe that further clarification by using the word “only” can serve to reduce ambiguity in the preregistration.” (p.23)
Reviewer 4, Comment 11	What exactly is meant with "parameterisation of the model"? My interpretation would be to the version used to describe the *same* model (e.g., a Beta distribution can be described using alpha and beta, but also using a mode and a	That is a good point, thank you. By “parameterization of the model”, we mean both the different variants of a single model, as well as the different expressions of a single variant of a model. We now clarify this within the revised manuscript:

	concentration; see https://en.wikipedia.org/wiki/Beta_distribution#Alternative_parameterizations), but I think the authors mean to refer to which parameters are included to make up *different* models (e.g., some versions of the GCM use a response scaling parameter whereas others don't). I think it is important to avoid this potential confusion. How can the appropriate number of samples be meaningfully set before the chain is run? At least partially, the number of samples seems to depend on data-dependent issues, such as convergence?	“Note that when we refer to the ‘‘model parameterisation”, we are referring to both the different potential variants within a single model (e.g., a diffusion model with or without between-trial variability in drift rate), as well as the different ways that the parameters can be expressed within the same variant of a model (e.g., a diffusion model with starting point estimated as an absolute value or as a value relative to the threshold). We believe that both of these aspects are important to specify within the preregistration, as variations in how the parameters are expressed could potentially lead to different results (e.g., due to changes in priors, which can influence Bayesian model selection Kass & Raftery, 1995; Vanpaemel, 2010).” (p.15) Regarding the number of samples used in the MCMC algorithm, we agree that convergence is more important than pre-specifying the number of samples. However, we think that researchers familiar with using specific MCMC methods for specific models likely have a good idea of both how long the sampler requires to converge and what number of samples are computationally feasible. Stating this within the preregistration could help to justify potential later deviations from the preregistered plan. For example, if the sampler fails to converge within a reasonable number of samples on numerous attempts, this might suggest that accurately estimating the posterior distributions of the specific model may be infeasible, and an alternative (perhaps simpler) model should be used instead. We now clarify this within a footnote in the revised manuscript: “Note that while the number of samples taken in the Markov-chain Monte Carlo algorithm does not necessarily need to be specified a-priori, as the important factor is whether the sampling algorithm converges on the posterior distribution, we believe that this level of specification is (1) knowledge that researchers experienced with specific models and
--	--	--

		specific sampling algorithms likely have, and (2) useful to help justify potential deviations from the preregistered model, such as in situations where the model does not converge within a reasonable number of samples.” (p.24)
Reviewer 4, Comment 12	Further, I think the preregistration included as an example is fine as is. However, to serve as an *exemplary* preregistration, fine maybe not be good enough. A few examples of where I think it could be improved: I realize not much can be done, given that it is preregistration, but the absence of a robustness analysis makes the example application a rather poor example.	We see the reviewer’s point, and we agree that our example application certainly does not constitute a “perfect” preregistration. However, we did not intend for our example application to serve as a perfect, exemplary representation of what all preregistrations should look like. Rather, our example application was intended to showcase our use of the template to answer a realistic research question within the area of reward rate optimality. Furthermore, as we have attempted to emphasise throughout the manuscript, we do not believe that preregistration is a “one size fits all” solution: different studies will have different preregistrations, and the “perfect” preregistration is likely an unrealistic goal (similar to the “perfect study”). However, we agree that this may not have been clear within the previous versions of the manuscript, and we have attempted to clarify the purpose of our example application within the revised manuscript: “Note that we do not intend for our example application to be seen as a perfect or exemplary instance of preregistration in model application. Rather, our example application is intended to serve as our realistic attempt to preregister a research question involving model application within the area of reward rate optimality using our proposed preregistration template. Importantly, we do not believe that preregistration is a “one size fits all” solution: the preregistration of different model application studies (or even different analyses of the same study) are likely to differ from one another in many respects, meaning that the concept of a perfect/exemplary preregistration may not be particularly helpful. However, we hope that our

		example application serves as a useful example for readers who wish to use our template to preregister their own model application studies.” (p.21)
Reviewer 4, Comment 13	The "Example Application Results" section should do a better job linking the reported BFs to the different hypotheses. The redundancy between 1.4 and 8.1 is confusing. Why is the test of H1 classified as "other analyses"?	Good point; we now explicitly state when we are looking at each analysis, and what hypothesis/hypotheses each analysis is testing. Regarding the redundancy between 1.4 and 8.1 in our preregistration document, we note in 8.1 that we're restating the hypotheses from before. We believe that it is useful to restate these hypotheses here as a refresher, as this section explains how they are being tested. Regarding the separation between "8.1: Statistical models" and "8.2: Other analyses", 8.1 is restricted to the quantitative inferences, whereas 8.2 provides the more qualitative analyses (i.e., a qualitative assessment of the trends in the posterior distributions over time).